# Enabling arbitrary inference in spatio-temporal dynamic systems: A physics-inspired perspective

**Yan Ge**[1], **Zhengyang Zhou**[1,2,*], **Qihe Huang**[1], **Yuxuan Liang**[3], **Yang Wang**[1,2,*]
[1]University of Science and Technology of China (USTC), Hefei, AnHui, China
[2]Suzhou Institute for Advanced Research, USTC, Suzhou, Jiangsu, China
[3]Hong Kong University of Science and Technology (Guangzhou), Guangzhou, GuangDong, China
{geyan, hqh}@mail.ustc.edu.cn, zzy0929@ustc.edu.cn,
yuxliang@outlook.com, angyan@ustc.edu.cn

## Abstract

Modern spatio-temporal learning techniques usually exploit sampled discrete observations to foresee the future. Actually, spatio-temporal dynamics are continuous and evolve continuously across time and space, thus modeling spatio-temporal dynamics in a continuous space can be a long-standing challenge. Existing deep learning architectures often fail to generalize to unseen regions and new graph topologies, while many physics-driven approaches are confined to Euclidean grids and poorly scale to complex graph structures. To address this gap, we propose PhySTA, a physics-inspired spatio-temporal learning framework designed for efficient and scalable arbitrary inference over graph-structured data. PhySTA integrates two key modules: (1) Continuous Operator-based Spectrum-Temporal Learning (CoSTL), which leverages a Graph-Time Fourier Neural Operator combined with Time-Gated Spectral Segmentation Perception to model continuous dynamics in operator space, and (2) Adaptive Multi-scale Interaction (AMI) that constructs multi-scale subgraphs and introduces node-edge coupled convolution to capture discrete interaction patterns and refine continuous predictions. By bridging operator learning with node-edge-graph interaction, PhySTA achieves both continuity-aware dynamic modeling and hierarchical interactive refinement. Extensive experiments across large-scale benchmarks demonstrate that PhySTA attains state-of-the-art accuracy while reducing computation cost and lowering parameter overhead.

## 1 Introduction

Spatio-temporal systems represent a class of complex dynamical processes whose states evolve jointly over space and time, underpinning critical applications such as urban traffic network optimization (Zhang et al., 2018), climate forecasting, and environmental monitoring (Chen et al., 2023). A fundamental challenge in their modeling lies in the discrepancy between discrete sensor observations and the underlying continuous dynamics, as shown in Fig. 1. Due to the sparse deployment of sensors and limited sampling frequency, the available data are inherently discrete, whereas the dynamics of real-world systems unfold over continuous spatio-temporal fields. This discrepancy complicates the reconstruction of system-level dynamics and restricts reliable inference in unobserved regions, ultimately limiting the applicability of spatio-temporal inference models.

From the perspective of spatial domains, spatio-temporal systems can be broadly categorized into Euclidean systems, exemplified by fluid velocity fields, and non-Euclidean systems, exemplified by traffic flow graphs. Neural operators (Lu et al., 2021; Li et al., 2021; Kovachki et al., 2021) have shown universal approximation capabilities in learning operator mappings over function spaces, enabling effective modeling of continuous dynamics. However, such methods are limited to Euclidean domains. In contrast, Graph Neural Networks (GNNs) have emerged as the mainstream approach

---

*Zhengyang Zhou and Yang Wang are the corresponding authors.

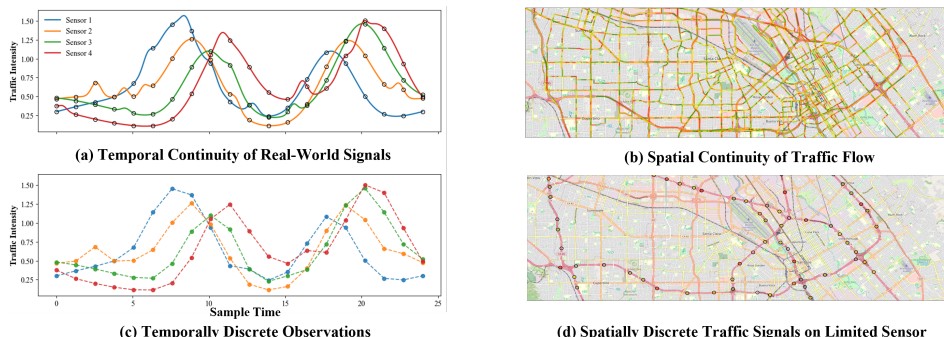

Figure 1: Figures (a) and (b) illustrate continuous real-world signals, whereas (c) and (d) depict their temporally and spatially discrete observations, emphasizing the challenge of bridging continuous gaps in both dimensions.

for *non-Euclidean spatio-temporal systems* (Li et al., 2018), encompassing static structural models (Guo et al., 2019), diffusion-based methods, and dynamic graph models (Bai et al., 2020; Wu et al., 2020). While these approaches capture node-level dynamics, they often struggle to explicitly model node-edge coupling within graph topologies and rely heavily on network depth to capture multi-scale interactions, leading to inefficiency and performance bottlenecks.

Taken together, existing studies still fail to simultaneously address continuous evolution modeling and discrete interaction learning, thereby limiting predictive accuracy and generalization to unseen regions. To overcome this gap, we identify two core tasks for continuous spatio-temporal modeling on graphs: (1) **graph neural operator construction and spectral modeling**, which extends the continuous modeling capacity of neural operators to graph-structured data and enables efficient spectral-mode learning for dynamic representations, and (2) **enhancement of graph neural interactions**, which refines operator-driven inference by capturing discrete multi-scale node-edge dynamics while alleviating the computational burden associated with deep GNN stacking.

Inspired by the physical principles of operator theory for continuous dynamical modeling (Karniadakis et al., 2021; Lu et al., 2021) and multi-body gravitational interactions (Shabana, 2020; Goldstein et al., 2002), we propose **PhySTA**, a *Physics-inspired Spatio-Temporal Learning framework for Arbitrary Inference*. PhySTA instantiates non-Euclidean systems as spatio-temporal graphs, and integrates the continuous modeling of neural operators with the discrete interaction adaptivity of graph neural networks, enabling efficient and generalizable arbitrary inferences.

Concretely, we first construct a *Graph-Time Fourier Neural Operator (GT-FNO)* based on the magnetic Laplacian, extending the neural operator paradigm to directed graphs. To balance accuracy and efficiency, we introduce *Time-Gated Spectral Segmentation Perception*, which adaptively assigns independent or shared parameters across spectral bands depending on stationarity, and applies time gating to dynamically reweight spectral components. For graph-structured interactions, we design a *Node-Edge Coupled Convolution* to capture the joint effects of node and edge features during message passing. Further, inspired by multi-grid solvers, we build a three-level coarse-mid-fine subgraph hierarchy that enables efficient multi-scale interaction learning within a single layer.

Our main contributions are summarized as follows:

- **Continuous Learning for Spatio-Temporal Systems**: GT-FNO extends neural operators to non-Euclidean domains, while Time-Gated Spectral Segmentation Perception learns spectral dynamics and temporal evolution, achieving universal approximation of graph-based continuous operators.

- **Node–Edge Coupled Multi-Scale Graph Convolution**: A unified mechanism that couples node and edge dynamics with multi-scale subgraphs, enabling single-layer modeling of cross-scale interactions and mitigating continuous inference errors.

- **Efficient Arbitrary Inference**: Comprehensive evaluation on large-scale traffic and air quality benchmarks demonstrates that PhySTA achieves state-of-the-art accuracy with significantly reduced computational cost (up to 74.6% fewer FLOPs) and parameters, effectively supporting decision-making under sparse sensor coverage.

## 2 RELATED WORK

**Euclidean Spatio-Temporal Modeling.** Classical approaches solve spatio-temporal dynamics by discretizing continuous Euclidean domains and numerically integrating physical equations. Representative methods include the finite difference method (FDM) (LeVeque, 2007), finite element (FEM) (Quarteroni, 2008), finite volume (FVM), and spectral methods (Canuto et al., 2007), which enable high-fidelity simulation but suffer from high computational cost, complex meshing, and entangled numerical and modeling errors. With the rise of deep learning, CNN-based architectures (Zhang et al., 2017; Ye et al., 2019) were proposed to capture spatial patterns on structured grids, reducing computational cost but remaining tied to regular meshes. Physics-informed neural networks (PINNs) (Raissi et al., 2019; Kashinath et al., 2021) embed PDE residuals into the training objective, supporting both forward and inverse problems without explicit meshing, though they require expensive auto-differentiation and retraining across PDE families. More recently, neural operators (Lu et al., 2021; Li et al., 2021; Kovachki et al., 2021) have been proposed to learn mappings between function spaces, enabling generalization across equations and resolutions. Representative examples include DeepONet, which employs a branch–trunk architecture to jointly encode function inputs and query coordinates, and the Fourier Neural Operator (FNO), which parameterizes global convolution kernels in the spectral domain. Despite these advances, these approaches remain largely confined to Euclidean spaces and face challenges in handling non-Euclidean domains such as graphs, as well as in mitigating multi-scale accumulation errors in large-scale continuous modeling.

**Non-Euclidean Spatio-Temporal Modeling.** In non-Euclidean domains, graph neural networks (GNNs) have become the dominant paradigm for capturing complex topological structures, with widespread applications in traffic forecasting and environmental monitoring. Early models, including STGCN (Yu et al., 2018), DCRNN (Li et al., 2018), GWNet (Wu et al., 2019), and AST-GCN (Guo et al., 2019), combine static graph convolutions with temporal encoders (e.g., recurrent units, diffusion modules, or attention) to capture spatio-temporal dependencies. To address evolving topologies, dynamic graph models have been developed: DGCRN (Li et al., 2023) jointly learns latent graphs and node dynamics, DSTAGNN (Lan et al., 2022) infers time-varying adjacency matrices, and D2STGNN (Shao et al., 2022b) disentangles spatial and temporal components via hierarchical modeling. More recent works emphasize temporal continuity and expressive power, such as AGCRN (Bai et al., 2020) with node-specific attention, STTN (Li & Zhu, 2021) combining Transformers with spatial convolution, and STG-ODE (Fang et al., 2021) reformulating graph convolution as continuous-time ODEs. Beyond GNNs, alternative paradigms have emerged, including generative modeling (Wen et al., 2023), multi-task learning (Yuan et al., 2024), domain transfer (Liu et al., 2025), and large-scale self-supervised pretraining (Zha et al., 2024; Ma et al., 2024). Nonetheless, they largely operate on discrete representations, leading to suboptimal performance in continuous non-Euclidean scenarios. Moreover, most graph-based methods focus on node propagation patterns while underrepresenting node–edge interactions, which constrains generalization in dynamic heterogeneous environments. This reliance on deep stacking for global pattern learning further hampers their compatibility with continuous modeling methods, exacerbating both accuracy and efficiency bottlenecks when addressing multi-scale error accumulation.

## 3 METHODOLOGY

### 3.1 PRELIMINARY

**Problem Statement.** Given a spatiotemporal dataset, we extract a temporal window of length $T$, $X_{1:T} = \{X_t\}_{t=1}^{T}$, where each $X_t \in \mathbb{R}^{N \times C}$ represents node features over a graph with adjacency matrix $A \in \mathbb{R}^{N \times N}$ and $C$-dimensional variables per node. Let $\mathcal{M}$ denote the continuous spatial manifold that includes both observed and unobserved locations.

Our goal is to learn a continuous operator $\Phi$ that, given the historical observations $X_{1:T}$ and graph structure $A$, predicts signal values at arbitrary spatial positions $s \in \mathcal{M}$ and future timestamps $t \in \{T+1, \ldots, T+\tau\}$:

$$\hat{x}(s,t) = \Phi(X_{1:T}, A; s, t), \tag{1}$$

where $\hat{x}(s,t)$ denotes the predicted signal at a (possibly unobserved) location $s$ and future time $t$.

## 3.2 THE OVERALL FRAMEWORK

In this section, we introduce **PhySTA**, a physics-inspired spatio-temporal learning framework designed for continuous modeling and arbitrary inference on graph-structured domains. At its core, PhySTA integrates three complementary components. First, the *Graph–Time Fourier Neural Operator* (GT-FNO) employs a *Time-Gated Spectral Segmentation Perception* to approximate the solution operator of the target spatio-temporal system in the spectral domain. Second, the *Adaptive Multi-scale Interaction* (AMI) module, inspired by multi-body dynamics, captures discrete node–edge coupled interactions across multiple scales to refine both global and local dependencies. Finally, Continuity–Discreteness Interaction Module (CDIM) unifies CoSTL for continuous inference and AMI for discrete refinement, supporting generalization to unseen regions and correcting long-horizon operator errors.

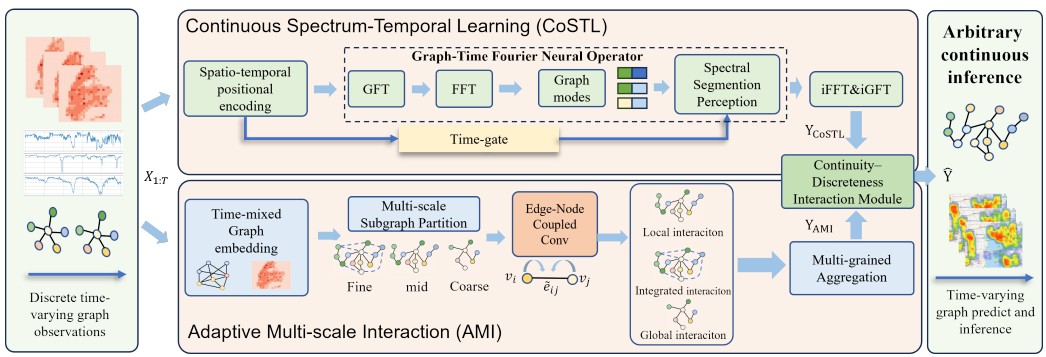

Figure 2: Overview of the PhySTA framework. Taking historical node observations $X_{1:T}$ and the adjacency matrix $A$ as inputs, the Continuity-Discreteness Interaction Module (CDIM) fuses the intermediate outputs $[Y_{\text{CoSTL}}, Y_{\text{AMI}}]$ to yield the final prediction $\hat{Y}$.

## 3.3 CONTINUOUS SPECTRUM-TEMPORAL MODELING

The paradigm of Fourier transforms with finite-dimensional mappings has been proven to enable continuous operator approximation learning on Euclidean spaces. GT-FNO extends this paradigm to graph domains by transforming spatio-temporal signals into the spectral space. To capture dynamics more effectively, *time-gated spectral segmentation perception* partitions modes into different frequency bands, while temporal gating modulates their evolution. The spatiotemporal embedding follows ST-FNO(Cao et al., 2024), explicitly providing high-frequency, multiscale spatiotemporal basis functions that help the network capture fine details and temporal evolution.

**Graph–Time Joint Spectral Decomposition.** Given a graph signal $X_{1:T}$, we first perform the Graph Fourier Transform (GFT) on the graph domain using the complex eigenvectors $\{\varphi_q\}$ of the magnetic Laplacian to encode directional dependencies (for undirected graphs we use the vanilla Laplacian):

$$X_{gft}(q,t) = \sum_{n=1}^{N} \varphi_q(n) \, X(n,t). \tag{2}$$

This transformation maps the representation from the node domain $(n, \cdot)$ to the graph spectral domain $(q, \cdot)$. Subsequently, a 1D Fast Fourier Transform (FFT) is applied along the temporal axis for each graph mode:

$$X_{gtft}(q,\omega) = \sum_{t=1}^{T} X_{gft}(q,t) \, e^{-i\omega t}. \tag{3}$$

This step converts the representation from the time domain $(q, t)$ to the temporal frequency domain $(q, \omega)$. Such a joint spectral embedding serves as the foundation for spatiotemporal operator construction.

**Time-Gated Spectral Segmentation Perception.** Let $\lambda \in \mathbb{R}^K$ denote the vector of Laplacian eigenvalues, where $K$ is the number of retained spectral modes. Each eigenvalue–eigenvector pair

corresponds to a graph mode in the spectral domain. Spectral modes are first partitioned into negative and positive sets:

$$\mathcal{I}_{\text{neg}} = \{k : \lambda_k < 0\}, \qquad \mathcal{I}_{\text{pos}} = \{k : \lambda_k \geq 0\}. \tag{4}$$

Within $\mathcal{I}_{\text{pos}}$, modes are further divided into low, mid, and high bands based on cumulative energy or predefined thresholds $s = (s_0, s_1)$ (e.g., $s = (0.1, 0.95)$), yielding $\mathcal{I}_{\text{low}}, \mathcal{I}_{\text{mid}}$, and $\mathcal{I}_{\text{high}}$.

GT-FNO applies differentiated parameterization across frequency bands. For low frequencies $\mathcal{I}_{\text{low}}$, each mode is assigned an independent kernel $W_k^{\text{low}}$ to preserve expressive capacity. Conversely, to capture fine-scale variations while controlling parameter complexity, modes within the negative, mid, and high frequency bands share a band-specific kernel ($W^{\text{neg}}, W^{\text{mid}}$, and $W^{\text{high}}$, respectively) scaled by an additional learnable per-mode factor $\alpha_k$. For a detailed analysis of the relationships between dynamical components and spectral bands, see Appendix A.4.

The spectral segmentation perception is expressed as:

$$X_{ssp}(k, \omega) = \begin{cases} W_k^{\text{low}} \cdot X_{gtft}(k, \omega), & k \in \mathcal{I}_{\text{low}}, \\ \alpha_k \cdot \left( W^{\text{band}(k)} X_{gtft}(k, \omega) \right), & k \in \mathcal{I}_{\text{neg}} \cup \mathcal{I}_{\text{mid}} \cup \mathcal{I}_{\text{high}}, \end{cases} \tag{5}$$

where $W^{\text{band}(k)} \in \{W^{\text{neg}}, W^{\text{mid}}, W^{\text{high}}\}$ corresponds to the kernel assigned to the specific frequency band containing mode $k$. This low-rank piecewise approximation preserves strong expressivity for long-term low-frequency trends while significantly reducing parameters in the mid and high frequencies, balancing stability and efficiency.

To enhance the modeling of non-stationary dynamics, we introduce a time-gating mechanism on the spectral perception outputs. Let $h(t)$ denote the absolute temporal embedding of the input sequence, which is transformed into the frequency domain $\tilde{h}(\omega)$ via 1D FFT for alignment. First, a gating factor $g(\omega)$ is generated based on this temporal embedding:

$$g(\omega) = \sigma(W_g \tilde{h}(\omega) + b_g), \tag{6}$$

where $\sigma$ denotes the Sigmoid activation function, and $W_g, b_g$ are learnable parameters. Subsequently, this gating factor is applied uniformly across the spectral segments to modulate the features:

$$X_{tgssp}(k, \omega) = g(\omega) \cdot X_{ssp}(k, \omega). \tag{7}$$

This allows adaptive modulation of each frequency band along temporal frequencies to better capture non-stationary evolution.

**Continuous Reconstruction.** After bandwise filtering and temporal gating, the processed spectral coefficients are mapped back to the spatiotemporal domain by sequential inverse transforms:

$$Y_{\text{CoSTL}}(n, t) = \sum_{k=1}^{K} \sum_{\omega=1}^{T} X_{tgssp}(k, \omega) \, \varphi_k(n) \, e^{i\omega t}, \tag{8}$$

where $\varphi_k$ denotes the eigenvectors of the magnetic Laplacian that capture graph structural harmonics, and $e^{i\omega t}$ represents the temporal Fourier basis modes.

The Fourier transform projects the input field onto continuous bases in the spectral domain, where a finite-dimensional truncation yields a computable set of spectral coefficients. The learned operator is then mapped back to the spatiotemporal domain via inverse transforms. Owing to the completeness of Fourier bases in $L^2$ spaces and their inherent ability to encode global dependencies, GT-FNO effectively learns continuous operator approximations, rather than discrete function mappings as in conventional neural networks. The approximation error primarily arises from spectral truncation and finite parameterization. To mitigate this, we introduce an Adaptive Multi-Scale Interaction (AMI) module, which models multi-scale discrete interactions and compensates for errors induced during continuous prediction. Detailed proofs of continuity and the algorithm process are provided in Appendices A.2 and A.3.

## 3.4 ADAPTIVE MULTI-SCALE INTERACTION

To mitigate the cumulative errors introduced by operator learning in continuous inference, we incorporate multi-scale discrete interactions through graph networks. Conventional graph convolution fixes interaction weights by static edges, limiting the modeling of dynamic and heterogeneous

dependencies. Inspired by gravitational physics, node–edge coupled convolution jointly encodes node features and edge attributes to generate adaptive interaction weights. For long-range dependencies, multi-scale subgraph module constructs coarse- and fine-scale subgraphs and fuses their convolutions, enabling efficient global–local interaction modeling within a single layer. The time-mixed embedding follows STID (Shao et al., 2022a), offering joint spatiotemporal basis features that strengthen the model's ability to capture evolving cross-time correlations.

**Adaptive Edge-Node Coupled Graph Convolution.**

For each edge $(i, j) \in E$ with attribute $e_{ij}$ and neighbor node feature $v_j \in \mathbb{R}^H$:

$$\tilde{e}_{ij} = \text{MLP}_e(e_{ij}) \in \mathbb{R}^H,$$
$$[\gamma_{ij}, \beta_{ij}] = \text{MLP}_\phi([\tilde{e}_{ij}; v_j]), \quad (9)$$

where $\gamma_{ij}, \beta_{ij} \in \mathbb{R}^H$ are FiLM (Feature-wise Linear Modulation) coefficients for a *node-conditioned* setting. Each edge dynamically modulates messages based on both edge attributes and neighboring node features.

During message passing, neighbor features are modulated as $m_{ij} = \gamma_{ij} \odot v_j + \beta_{ij}$, and node states are updated via:

$$v_i' = \sigma\Big( \sum_{j \in \mathcal{N}(i)} m_{ij} \Big), \quad (10)$$

Figure 3: Edge-node coupled graph convolution. Edge attributes and node features jointly produce parameters $[\gamma_{ij}, \beta_{ij}]$ to modulate messages $m_{ij}$, which are aggregated to update node representations $v_i'$.

where $\sigma$ is an activation function (e.g., ReLU).

Because $v_j$ encodes local or historical context, the FiLM parameters $\gamma_{ij}$ and $\beta_{ij}$ evolve dynamically, endowing the convolution with *spatiotemporal adaptivity*. This design preserves edge-driven influences while incorporating node features, achieving heterogeneous, anisotropic, and context-sensitive message modulation.

**Multi-scale Single-Layer Long-Range Modeling.** The multi-scale subgraph module begins by applying community detection to partition the original graph. A coarse graph, composed of induced nodes (community-level abstractions), is used to capture cross-community interactions; the middle graph preserves local interactions; and a fine graph, obtained by superimposing community edges onto the original graph, models global interactions. By integrating node-edge coupled convolutions across these coarse, middle, and fine subgraphs, the module enables simultaneous learning of global trends, regional dependencies, and local refinements within a single forward pass.

Let the original adjacency matrix be $A \in \mathbb{R}^{N \times N}$, clipped to non-negative values for metric consistency. We maximize modularity, $Q = \frac{1}{2m} \sum_{i,j} \left( A_{ij} - \frac{k_i k_j}{2m} \right) \delta(c_i, c_j)$ via Louvain community detection and select the highest-degree node in each community as an inducing point. This defines a mapping $\text{node2center}[i] = u$ for every $i \in C_u$.

We capture global, local, and integrated propagation patterns via a three-scale graph design. At the **coarse scale**, which models global trends, each community selects a high-order representative node. The original edge weights $A_{ij}$ are aggregated into induced edges $A_{\text{coarse}}[u, v]$, and an edge-node dynamic convolution on this downsampled edge set $E_{\text{down}}$ produces the coarse-scale features $X_{\text{coarse}}$. At the **mid scale**, designed to capture local interactions, we strictly preserve the original adjacency ($A_{\text{mid}} = A$) and apply the same convolution on $E_{\text{mid}}$ to extract mid-scale features $X_{\text{mid}}$. Finally, at the **fine scale**, which provides integrated refinement, the coarse edge weights are redistributed back to the original node pairs. These weights are then sparsified via a Top-K operator and combined with the original adjacency to form $A_{\text{fine}}$. A final convolution on $E_{\text{fine}}$ yields the fine-scale features $X_{\text{fine}}$, seamlessly merging global semantics with local structural details.

**Multi-scale aggregation.** Finally, we concatenate the original features with the representations from coarse, mid, and fine scales:

$$Y_{\text{AMI}} = \text{Concat}(X_{1:T}, X_{\text{coarse}}, X_{\text{mid}}, X_{\text{fine}}) \in \mathbb{R}^{N \times (C+3H)}. \quad (11)$$

This unified representation captures a comprehensive spectrum of graph signal behaviors—from low-frequency global trends to high-frequency localized variations. The resulting three-scale structure $\{A_{\text{coarse}}, A_{\text{mid}}, A_{\text{fine}}\}$, integrated with node-edge dynamic convolution, enables *hierarchical, cross-scale, and tunable* spatial modeling in a single layer. It *down-aggregates* long-range dependencies, *smooths* intermediate interactions, and *up-refines* global semantics—offering strong utility for large-scale heterogeneous graph prediction.

### 3.5 ARBITRARY INFERENCE

Finally, our model integrates the two complementary outputs. The initial prediction from the Time-Gated Spectral Segmentation Perception, $Y_{\text{CoSTL}}$, represents *unconstrained extrapolation*, while the correction signal from the AMI module, $Y_{\text{AMI}}$, encodes *adaptive refinement based on known observations*. These outputs are concatenated and fused via the proposed Continuity–Discreteness Interaction Module (CDIM):

$$\widehat{Y} = \text{CDIM}\big([Y_{\text{CoSTL}}, Y_{\text{AMI}}]\big). \tag{12}$$

Specifically, CDIM is implemented as a multi-layer perceptron, $\text{CDIM}(z) = W_2\big(\text{Dropout}(\phi(W_1 z + b_1))\big) + b_2$, where $W_1 \in \mathbb{R}^{d_{\text{in}} \times d_{\text{emb}}}$ and $W_2 \in \mathbb{R}^{d_{\text{emb}} \times d_{\text{out}}}$, $\phi$ is a nonlinear activation function, and $z = [Y_{\text{CoSTL}}, Y_{\text{AMI}}]$ concatenates the coarse predictions and adaptive refinements.

Unlike simple gating, CDIM provides higher flexibility by learning nonlinear correction functions (Hornik et al., 1989), thereby avoiding the fragmentation of predictions between continuous extrapolation and discrete correction. Together, this formulation reflects a cohesive *coarse-to-fine reasoning path*: **CoSTL** performs structure-aware spectral decomposition and inverse transformation to generalize across arbitrary target nodes in continuous space, while **AMI** adaptively refines these predictions through multiscale node–edge dynamic convolution over a hierarchical graph.

## 4 EXPERIMENTS

In this section, we present extensive experiments conducted on three real-world datasets to evaluate the proposed **PhySTA** across various levels of inference and prediction tasks.

### 4.1 EXPERIMENTAL SETTINGS

**Datasets.** We evaluate our model on three real-world benchmarks for traffic and air quality forecasting: PEMS-BAY (Chen et al., 2001), SD (Liu et al., 2023), and KnowAir (Wang et al., 2020). PEMS-BAY contains 325 sensor readings of highway speeds in the California Bay Area sampled every 5 minutes, capturing rush hours, holidays, and incidents. SD is a subset of LargeST, spanning diverse urban regions with heterogeneous transportation structures (highways and local roads). KnowAir provides hourly air quality data from dense monitoring networks, including pollutants (e.g., $PM_{2.5}$, $NO_2$) and meteorological covariates (e.g., wind, temperature). All datasets are chronologically split into training, validation, and test sets under a strict 6:2:2 ratio (Li et al., 2018), ensuring no temporal leakage.

**Baselines.** We compare PhySTA against representative approaches from three categories: (1) *Static graph models*: STGCN (Yu et al., 2018), GWNet (Wu et al., 2019), ASTGCN (Guo et al., 2019); (2) *Dynamic graph models*: DGCRN (Li et al., 2023), DSTAGNN (Lan et al., 2022), D2STGNN (Shao et al., 2022b); (3) *ODE-/attention-based models*: AGCRN (Bai et al., 2020), STTN (Li & Zhu, 2021), and STG-ODE (Fang et al., 2021). These cover mainstream paradigms of spatio-temporal forecasting. We adopt the best-performing configurations and official implementations of each baseline for fair comparison.

**Evaluation Metrics.** Following prior work, we report *Mean Absolute Error (MAE)*, *Root Mean Squared Error (RMSE)*, and *Mean Absolute Percentage Error (MAPE)*. MAE captures average deviation, RMSE emphasizes large errors, and MAPE provides percentage-based interpretability. Together, these metrics offer a comprehensive evaluation of forecasting accuracy and robustness.

---

```
https://github.com/liyaguang/DCRNN
https://www.kaggle.com/datasets/liuxu77/largest
https://github.com/shuowang-ai/PM2.5-GNN
```

Table 1: Inference performance across different datasets under varying missing rates. The best and second-best results are highlighted in **bold** and underlined, respectively.

| Dataset | Method | Mask=0 | | | Mask=0.3 | | | Mask=0.5 | | | Mask=0.7 | | |
|---|---|---|---|---|---|---|---|---|---|---|---|---|---|
| | | MAE | MAPE | RMSE | MAE | MAPE | RMSE | MAE | MAPE | RMSE | MAE | MAPE | RMSE |
| KnowAir | AGCRN | 21.38 | 0.60 | 33.63 | 27.11 | 0.91 | 39.61 | 31.94 | 1.15 | 44.39 | 36.91 | 1.39 | 48.58 |
| | ASTGCN | 21.67 | 0.55 | 34.96 | 27.36 | 0.85 | 40.41 | 31.02 | 1.10 | 42.63 | 31.47 | 1.15 | 42.64 |
| | DGCRN | 42.04 | 1.27 | 59.21 | 39.53 | 1.33 | 52.26 | 43.69 | 1.58 | 57.16 | 51.37 | 1.90 | 64.04 |
| | GWNET | 20.70 | 0.57 | 33.19 | 24.41 | 0.75 | 37.63 | 27.45 | 0.90 | 40.12 | 30.87 | 1.07 | 43.94 |
| | LSTM | 21.87 | 0.61 | 34.60 | 25.05 | 0.74 | 39.78 | 27.18 | 0.84 | 42.25 | 29.05 | 0.90 | 45.02 |
| | STGCN | 20.60 | 0.55 | 32.97 | 24.69 | 0.76 | 37.51 | 27.36 | 0.89 | 40.87 | 29.75 | 0.97 | 44.23 |
| | STGODE | 21.55 | 0.52 | 35.38 | 26.41 | 0.85 | 39.21 | 28.06 | 0.96 | 40.87 | 29.19 | 0.95 | 44.34 |
| | STTN | 20.99 | **0.49** | 35.30 | 23.68 | 0.67 | 37.56 | 26.84 | 0.86 | 40.10 | 29.37 | 1.01 | **41.91** |
| | PHYSTA | **20.55** | 0.55 | 33.05 | **22.89** | **0.65** | **36.63** | **25.20** | **0.74** | **39.84** | **27.19** | **0.82** | 42.58 |
| PEMS-BAY | AGCRN | **1.64** | 0.04 | 3.64 | 2.86 | 0.07 | 6.50 | 3.60 | 0.09 | 7.85 | 4.39 | 0.12 | 8.96 |
| | ASTGCN | 1.89 | 0.04 | 4.03 | 3.86 | 0.09 | 7.40 | 6.53 | 0.13 | 10.23 | 5.44 | 0.12 | 8.79 |
| | DGCRN | 1.85 | 0.04 | 4.08 | 2.97 | 0.07 | 6.13 | 3.70 | 0.09 | 7.19 | 4.40 | 0.11 | **8.14** |
| | GWNET | 1.65 | 0.04 | 3.63 | 2.97 | 0.07 | 6.22 | 3.64 | 0.09 | 7.26 | 4.43 | 0.11 | 8.23 |
| | LSTM | 1.83 | 0.04 | 4.13 | 2.98 | 0.07 | 6.40 | 3.78 | 0.09 | 7.39 | 4.54 | 0.11 | 8.36 |
| | STGCN | 1.82 | 0.04 | 3.91 | 3.26 | 0.08 | 6.37 | 4.09 | 0.10 | 7.49 | 4.98 | 0.12 | 8.42 |
| | STGODE | 1.71 | 0.04 | 3.72 | 2.92 | 0.07 | 6.16 | 3.72 | 0.09 | 7.47 | 4.42 | 0.11 | 8.43 |
| | STTN | 1.65 | 0.04 | 3.66 | 2.80 | 0.07 | 6.08 | 3.69 | 0.09 | 7.28 | 4.34 | 0.11 | 8.22 |
| | PHYSTA | 1.66 | **0.04** | **3.61** | **2.75** | **0.07** | **5.85** | **3.52** | **0.09** | **7.04** | **4.25** | **0.11** | 8.19 |
| SD | AGCRN | 21.24 | 0.17 | 34.90 | 64.24 | 1.11 | 110.73 | 87.96 | 1.54 | 130.37 | 116.23 | 2.15 | 154.52 |
| | ASTGCN | 28.18 | 0.21 | 43.05 | 81.89 | 1.40 | 153.63 | 108.81 | 1.79 | 162.41 | 120.48 | 1.75 | 166.48 |
| | DGCRN | 40.66 | 0.37 | 57.39 | 68.80 | 0.76 | 101.63 | 82.56 | 0.89 | 117.54 | 110.09 | 1.19 | 145.97 |
| | GWNET | 21.74 | 0.15 | 34.30 | 58.21 | 0.79 | 98.25 | 79.97 | 1.17 | 120.50 | 106.50 | 1.54 | 145.52 |
| | LSTM | 29.10 | 0.18 | 45.37 | 96.91 | 1.08 | 113.23 | 106.09 | 1.89 | 153.93 | 113.53 | 1.67 | 154.56 |
| | STGCN | 22.33 | 0.17 | 36.61 | 60.58 | 0.80 | 103.35 | 86.03 | 1.21 | 130.27 | 111.05 | 1.67 | 152.14 |
| | STGODE | 23.41 | 0.16 | 37.34 | 62.87 | 0.94 | 106.66 | 89.07 | 1.45 | 134.08 | 117.69 | 1.38 | 170.66 |
| | STTN | 22.40 | 0.15 | 34.85 | 55.73 | 0.76 | 92.03 | 82.95 | 1.34 | 122.23 | 108.46 | 1.75 | 144.34 |
| | PHYSTA | **20.64** | **0.15** | **33.05** | **50.34** | **0.45** | **91.65** | **77.64** | 1.06 | 119.43 | **96.09** | 1.40 | **134.59** |

## 4.2 PERFORMANCE COMPARISON

To evaluate the inference ability of PhySTA, we simulate unobserved regions during training by randomly masking a subset of node features. At test time, the same nodes are masked in the input while ground-truth labels remain fully available, enabling fair assessment of prediction under incomplete observations. Table 1 reports the MAE, MAPE, and RMSE averaged over 12 forecasting horizons.

From Table 1, PhySTA consistently achieves the best or second-best results across datasets and missing ratios. On KnowAir, it attains the lowest MAE under all masking conditions, outperforming strong baselines such as STTN and STGCN, especially at high sparsity (Mask = 0.7). On PEMS-BAY, PhySTA maintains stable accuracy even with severe missingness (Mask = 0.3, MAE = 2.75, RMSE = 5.85), surpassing advanced graph-based models including AGCRN and DGCRN. On the large-scale SD dataset, it demonstrates superior robustness at low-to-moderate missing ratios, reducing RMSE by over 5% compared to the next-best baseline while remaining competitive under extreme sparsity. In contrast, traditional models (e.g., LSTM, STGCN) degrade substantially at higher missing ratios, and advanced GNN baselines (e.g., DGCRN, STGODE) show fluctuating performance under sparse inputs. Overall, PhySTA's physics-guided spectral operator and adaptive multi-scale interaction enable stable, accurate predictions across diverse spatiotemporal patterns and sparsity levels, confirming its robustness and generalization for arbitrary inference in real-world spatiotemporal systems.

**Efficiency Analysis.** We theoretically derive the computational complexity of PhySTA and compare its parameter count and GPU memory consumption against representative baselines. As shown in Table 2, the time complexity of PhySTA is $O(d^2 nL)$, where $d$ denotes the feature dimension, $n$ is the number of graph nodes, and $L$ represents the number of integration steps (typically $L < n$). From Table 2, we observe that PhySTA uses only 123,474 parameters and requires 6,042MB of GPU memory, achieving a favorable trade-off between computational complexity and resource consumption. This demonstrates

Table 2: Efficiency comparison of models.

| Model | Complexity | Parameters | GPU memory (MB) |
|---|---|---|---|
| AGCRN | $O(d^2 nL)$ | 760,580 | 11,140 |
| ASTGCN | $O(d^2 nL)$ | 2,153,034 | 11,028 |
| DGCRN | $O(d^2 nL)$ | 242,849 | 15,058 |
| GWNET | $O(d^2 nL)$ | 311,164 | 9,274 |
| LSTM | $O(d^2 L)$ | 97,932 | 9,864 |
| STGCN | $O(d^2 nL)$ | 507,532 | 2,100 |
| STTN | $O(dn^2 L)$ | 113,740 | 18,864 |
| STGODE | $O(d^2 nL)$ | 729,228 | 18,864 |
| PhySTA | $O(d^2 nL)$ | 123,474 | 6,042 |

that integrating physics-inspired continuous modeling with multiscale message passing yields substantial performance gains within an acceptable memory footprint.

### 4.3 ABLATION STUDY

We perform a component-wise ablation study on the PEMS-BAY dataset under two masking ratios (`Mask = 0` and `Mask = 0.5`) to evaluate the contribution of key modules—Time-Gated Spectral Segmentation Perception (TGSSP), GT-FNO, Edge-Node Coupling Convolution (ENCC), Multi-Scale Graph Convolution Network (MSGCN), and the Multi-scale Interaction Refinement module (AMI).

Table 3: Ablation study of prediction(mask-0) and inference(mask=0.5) ability on PEMS-BAY.

| Methods | Mask = 0 | | | Mask = 0.5 | | |
|---|---|---|---|---|---|---|
| | MAE | MAPE | RMSE | MAE | MAPE | RMSE |
| Our Model | **1.66** | **0.04** | **3.61** | **3.52** | **0.09** | **7.04** |
| *w/o TGSSP* | 1.80 | 0.05 | 3.91 | 4.12 | 0.11 | 8.24 |
| *w/o GTFNO* | 2.05 | 0.06 | 4.46 | 5.03 | 0.14 | 10.06 |
| *w/o ENCC* | 1.73 | 0.04 | 3.76 | 3.87 | 0.10 | 7.74 |
| *w/o MSGCN* | 1.79 | 0.05 | 3.89 | 3.96 | 0.11 | 7.92 |
| *w/o AMI* | 1.88 | 0.05 | 4.09 | 4.13 | 0.12 | 8.26 |

As shown in Table 3, removing TGSSP (`w/o TGSSP`) leads to a minor drop: MAE increases from 1.66 to 1.80 and 3.52 to 4.12, with corresponding MAPE rising from 0.04 to 0.05 and 0.09 to 0.11. Excluding GT-FNO (`w/o GTFNO`) causes more pronounced degradation, with MAE reaching 2.05 and 5.03, and MAPE 0.06 and 0.14, highlighting its role in capturing spatio-temporal dynamics. Without ENCC (`w/o ENCC`), MAE increases to 1.73 and 3.87, and MAPE to 0.04 and 0.10, confirming the benefit of explicit edge-node coupling. Omitting MSGCN (`w/o MSGCN`) further reduces performance, particularly under `Mask = 0.5` (MAE = 3.96, MAPE = 0.11), showing the necessity of multi-scale sub-graph modeling.

Finally, removing AMI (`w/o AMI`) results in the largest drop, with MAE rising to 1.88 and 4.13, and MAPE to 0.05 and 0.12, demonstrating that adaptive multi-scale interaction refinement is crucial for robust prediction under incomplete data. A detailed hyperparameter sensitivity analysis for CoSTL and AMI (KnowAir, PEMS-BAY) is provided in Appendix A.7.

### 4.4 CASE STUDY

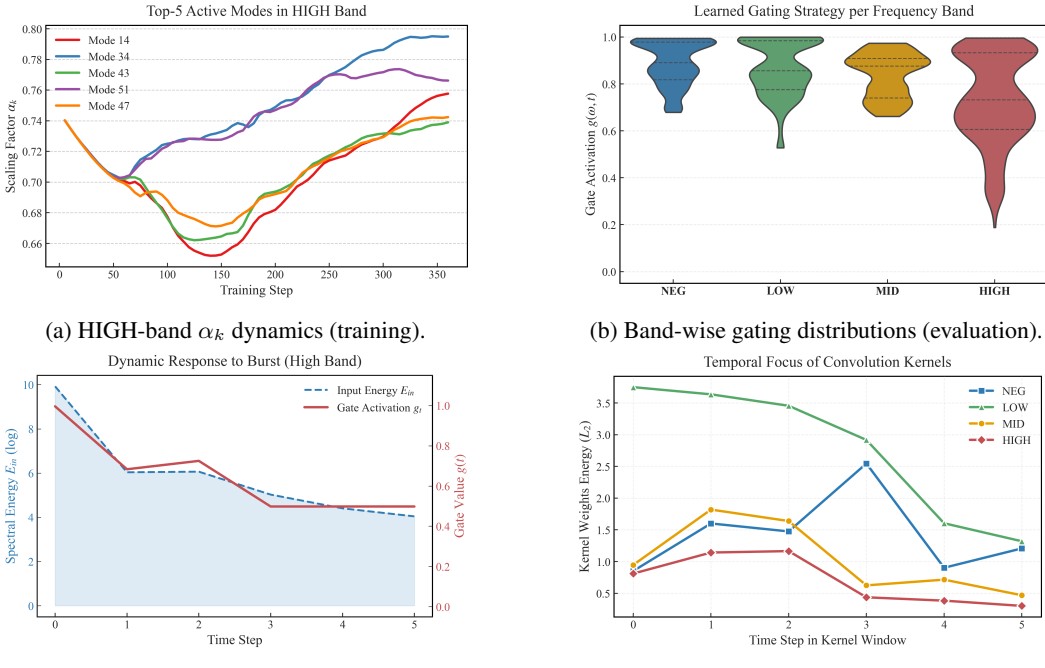

(a) HIGH-band $\alpha_k$ dynamics (training).

(b) Band-wise gating distributions (evaluation).

(c) Dynamic gating response (evaluation).

(d) Temporal kernel focus (evaluation).

Figure 4: Visualization of TGSSP's spectral specialization and temporal adaptation mechanisms.

**Spectral Specialization and Temporal Adaptation.** To dissect the internal learning mechanism of TGSSP, we analyze its two key parameter groups—*mode-wise scaling factors* ($\alpha_k$) and *temporal*

*gating activations*—and examine how they jointly shape spectral–temporal behavior across training and inference. As shown in Figure 4(a), which visualizes trajectories during training, $\alpha_k$ exhibits a progressive specialization effect: informative high-band modes are consistently amplified while less relevant ones decay, revealing a competitive "spectral selection" process. In contrast, Figures 4(b–d), derived from the evaluation phase, expose the model's stabilized temporal modulation strategy. The gating distributions in Figure 4(b) show that low-frequency modes adopt a nearly uniform pass-through policy to preserve global trends, whereas high-frequency modes exhibit a high-variance selective pattern tuned for local irregularities. This behavior aligns with the dynamic response in Figure 4(c), where gate activations spike synchronously with transient spectral bursts and decay as the signal stabilizes, effectively functioning as a spectral-domain attention mechanism that enhances signal–noise contrast. Finally, the temporal kernel profiles in Figure 4(d) indicate differentiated temporal receptive fields across frequency bands, confirming that TGSSP learns band-specific temporal focus indispensable for capturing multi-scale, non-stationary dynamics.

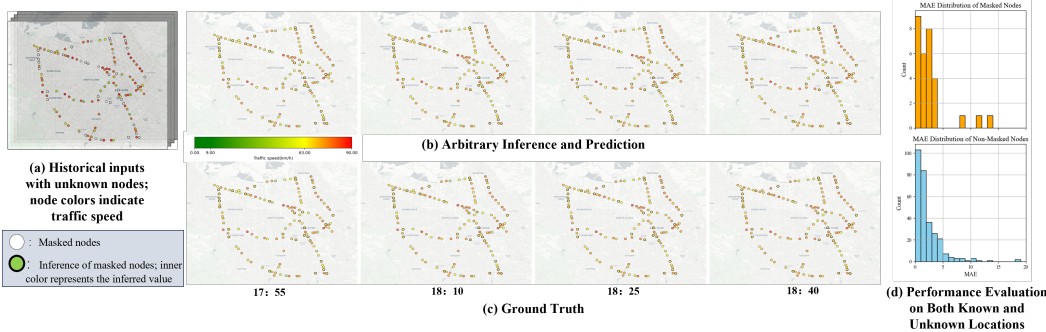

Figure 5: Visualization of PhySTA's spatiotemporal prediction and inference capabilities on transportation datasets.

**Arbitrary Spatiotemporal Inference.**  To demonstrate the arbitrary inference capability of PhySTA, we randomly mask 30% of the sensors, requiring the model to infer and predict traffic flow at both observed and unobserved locations. Specifically, the input spans the time interval from 16:50 to 17:40, during which 30% of the nodes are masked and treated as unknown regions. These unobserved nodes are visualized as white circles on the map. The model is then tasked with forecasting traffic flow for the subsequent period, from 17:55 to 18:40. As evidenced by the visualization comparison and the quantitative MAE evaluation against the ground truth, PhySTA accurately captures the underlying traffic dynamics and successfully performs arbitrary spatiotemporal inference over non-Euclidean domains, demonstrating strong generalization under partial observability.

## 5 Conclusion and Limitations

In this work, we proposed **PhySTA**, a physics-inspired framework that integrates a Graph–Time Fourier Neural Operator with an adaptive multi-scale interaction module. PhySTA enables continuous dynamics modeling on graph-structured data and performs multi-scale error correction, ultimately supporting arbitrary-region inference. Experiments on real-world benchmarks for traffic forecasting and air quality monitoring demonstrate that PhySTA not only outperforms prior methods but also achieves significant improvements in runtime efficiency and memory usage. Even under highly sparse sensor deployments, PhySTA can perform cross-region arbitrary inference, making it particularly suitable for real-time prediction and decision support at both city and national scales.

**Limitations and Future Work.**  Despite these promising results, the GT-FNO component still relies on spectral decompositions (e.g., magnetic Laplacian), which can be computationally demanding on very large graphs. Future work could explore graph-adaptive compression or cross-modal alignment techniques to mitigate these issues. Further, the scalability and generalizability of PhySTA to broader spatio-temporal domains, such as climate systems, power grids, or epidemiology, are highly required and remain as future work.

## 6 ETHICS STATEMENT

This work adheres to the ICLR Code of Ethics (https://iclr.cc/public/CodeOfEthics). Our research focuses on the development of graph-time neural operators for spatiotemporal modeling and does not involve human subjects, sensitive personal data, or experiments with potential direct harm. All datasets used in our experiments are publicly available or have been anonymized, and no identifiable information is included. We have considered potential biases in data and model evaluation, and all results are reported transparently. There are no conflicts of interest or external sponsorship that could influence the research outcomes. We encourage readers to consider ethical implications in the deployment of predictive models on real-world spatiotemporal data, particularly in sensitive applications such as urban monitoring or transportation, where fairness and privacy must be preserved.

## 7 REPRODUCIBILITY STATEMENT

We have made every effort to ensure the reproducibility of all experiments presented in this work. All model architectures, hyperparameters, and training procedures are described in detail in the main text and the appendix. All datasets used in our experiments are publicly available, and we provide links to their sources.

## 8 ACKNOWLEDGMENT

This paper is partially supported by the National Natural Science Foundation of China (Nos. 62502488, 12227901, and 62402414), the Natural Science Foundation of Jiangsu Province (No. BK20240460), the grant from the State Key Laboratory of Resources and Environmental Information System, and the Guangdong Basic and Applied Basic Research Foundation (No. 2025A1515011994). The AI-driven experiments, simulations, and model training were performed on the robotic AI-Scientist platform of the Chinese Academy of Sciences.

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

# A   APPENDIX

## A.1   NOTATION

Table 4: Frequently Appeared Notations

| Symbol | Description |
|---|---|
| $N, T, d$ | Number of sensors (nodes), historical time steps, and latent width. |
| $X_{1:T}$ | Historical spatio-temporal observation tensor. |
| $\mathcal{G}$ | Graph of spatial relationships. |
| $A$ | Weighted adjacency matrix. |
| $\Phi$ | Continuous solution operator / Graph spectral basis matrix. |
| $\widehat{Y}$ | Final fused spatio-temporal prediction. |
| $\varphi_k, \lambda_k$ | Graph Laplacian eigenvector and eigenvalue for mode $k$. |
| $X_{\text{gft}}(k, t)$ | Graph Fourier Transform representation. |
| $X_{\text{gtft}}(k, \omega)$ | Joint graph-time Fourier Transform representation. |
| $\mathcal{I}_{\text{neg}}, \mathcal{I}_{\text{pos}}$ | Negative and non-negative spectral mode sets. |
| $\mathcal{I}_{\text{low}}, \mathcal{I}_{\text{mid}}, \mathcal{I}_{\text{high}}$ | Low, mid, and high-frequency spectral bands. |
| $s = (s_0, s_1)$ | Band energy thresholds. |
| $W_k^{\text{low}}, W^{\text{band}(k)}$ | Mode-independent and band-shared learnable spectral weights. |
| $\alpha_k$ | Mode-wise scaling factor for negative, mid, and high bands. |
| $X_{\text{ssp}}(k, \omega)$ | Spectral segmentation perception output. |
| $h(t), \tilde{h}(\omega)$ | Temporal embedding and its frequency-domain representation. |
| $g(\omega)$ | Time-gating factor. |
| $X_{\text{tgssp}}(k, \omega)$ | Time-gated spectral output. |
| $Y_{\text{CoSTL}}(n, t)$ | Reconstructed continuous spatio-temporal output from CoSTL. |
| $e_{ij}, \tilde{e}_{ij}$ | Raw edge attribute and its projected embedding. |
| $\gamma_{ij}, \beta_{ij}$ | FiLM scaling and shifting coefficients for edge $(i, j)$. |
| $m_{ij}$ | Modulated message from neighbor $j$ to node $i$. |
| $X_{\text{coarse}}, X_{\text{mid}}, X_{\text{fine}}$ | Coarse, mid, and fine-scale node features from subgraphs. |
| $Y_{\text{AMI}}$ | Aggregated multi-scale interaction representation. |

## A.2   GT-FNO CONTINUOUS OPERATOR APPROXIMATION ON GRAPHS

We provide a formal justification for GT-FNO's ability to learn continuous operators on graph-structured spatio-temporal systems. Let $X : \mathcal{G} \times [0, T] \to \mathbb{R}^F$ denote a spatio-temporal graph signal defined on a directed graph $\mathcal{G}$ with $N$ nodes, where $[0, T]$ is the temporal domain. The target evolution operator is a continuous mapping $\mathcal{K} : L^2(\mathcal{G} \times [0, T]) \to L^2(\mathcal{G} \times [0, T])$. We aim to show that GT-FNO can approximate $\mathcal{K}$ arbitrarily well.

**1. Spectral Decomposition on Graphs.** Let $\mathbf{L}$ denote the magnetic Laplacian of $\mathcal{G}$ and $\{\varphi_k\}_{k=1}^N$ its orthonormal complex eigenvectors with eigenvalues $\{\lambda_k\}$. Any graph signal $X(\cdot, t)$ admits a spectral decomposition:

$$X(n, t) = \sum_{k=1}^N \hat{X}(k, t)\varphi_k(n), \quad \hat{X}(k, t) = \langle X(\cdot, t), \varphi_k \rangle_{\ell^2}.$$

Similarly, temporal signals are expanded via Fourier series:

$$\hat{X}(k, t) = \sum_{\omega \in \Omega_T} \tilde{X}(k, \omega)e^{i\omega t},$$

where $\Omega_T$ is the discretized temporal frequency set. Combining these, any graph-time signal admits a joint graph-time spectral decomposition:

$$X(n, t) = \sum_{k=1}^N \sum_{\omega \in \Omega_T} \tilde{X}(k, \omega)\varphi_k(n)e^{i\omega t}.$$

**2. Operator Approximation in the Spectral Domain.** The target operator $\mathcal{K}$ acts on $X$ as

$$(\mathcal{K}X)(n, t) = Y(n, t).$$

Using the spectral decomposition, we have:

$$Y(n, t) = \sum_{k=1}^{N} \sum_{\omega \in \Omega_T} \tilde{Y}(k, \omega) \varphi_k(n) e^{i\omega t}, \quad \tilde{Y}(k, \omega) = \mathcal{F}[\mathcal{K}]\big(\tilde{X}(k, \omega)\big),$$

where $\mathcal{F}[\mathcal{K}]$ denotes the induced action of $\mathcal{K}$ in the joint graph-time frequency domain.

**3. Low-Rank Piecewise Approximation via GT-FNO.** GT-FNO approximates $\mathcal{F}[\mathcal{K}]$ using a piecewise spectral mapping:

$$\tilde{U}(k, \omega) = \begin{cases} W_k^{\text{low}} \tilde{X}(k, \omega), & k \in \mathcal{I}_{\text{low}}, \\ \alpha_k\big(W^{\text{band}(k)} \tilde{X}(k, \omega)\big), & k \notin \mathcal{I}_{\text{low}}, \end{cases}$$

where $W^{\text{band}(k)}$ is the shared kernel assigned to the specific frequency band (mid, high, or negative) containing mode $k$. This is equivalent to approximating $\mathcal{F}[\mathcal{K}]$ by a finite-dimensional linear operator in each frequency band, which by standard results in functional analysis constitutes a spectral Galerkin approximation:

$$\|\mathcal{F}[\mathcal{K}] - \mathcal{F}[\mathcal{K}]_{\text{GT-FNO}}\|_{L^2} \leq \epsilon(N_{\text{low}}, N_{\text{mid}}, N_{\text{high}}),$$

where $\epsilon$ vanishes as the number of modes in each band increases.

**4. Universal Approximation of Continuous Operators.** Based on the universal approximation theorem for Fourier Neural Operators (Kovachki et al., 2021), let $\mathcal{K} : L^2(\mathcal{G} \times [0, T]) \to L^2(\mathcal{G} \times [0, T])$ be a continuous operator. For any $\epsilon > 0$, there exists a piecewise linear Fourier operator $\mathcal{K}_\theta$ such that

$$\|\mathcal{K}X - \mathcal{K}_\theta X\|_{L^2} < \epsilon, \quad \forall X \in \mathcal{B}_R,$$

where $\mathcal{B}_R$ is a bounded subset of $L^2$. Since GT-FNO implements exactly such a bandwise linear spectral operator, augmented by time-gating which is Lipschitz continuous, it inherits the universal approximation property on graph-time signals:

$$\forall \epsilon > 0, \quad \exists \theta : \|\mathcal{K}X - \text{GT-FNO}_\theta(X)\|_{L^2} < \epsilon.$$

**5. Error Bounds.** The total approximation error consists of three components:

$$\|\mathcal{K}X - \text{GT-FNO}_\theta(X)\|_{L^2} \leq \underbrace{\|\mathcal{K}X - \mathcal{K}_N X\|_{L^2}}_{\text{spectral truncation}} + \underbrace{\|\mathcal{K}_N X - \mathcal{K}_\theta X\|_{L^2}}_{\text{parameterization error}} + \underbrace{\|\mathcal{K}_\theta X - \text{GT-FNO}_\theta(X)\|_{L^2}}_{\text{time-gating approximation}}.$$

All three terms can be made arbitrarily small by increasing the number of graph/temporal modes and the capacity of the parameterized linear maps, guaranteeing continuous operator approximation on graph-time domains.

**6. Conclusion.** GT-FNO provides a theoretically grounded continuous operator approximation framework on graphs, extending the universality of FNOs to graph-structured spatio-temporal data. By leveraging spectral decomposition, bandwise linear parameterization, and Lipschitz continuous time-gating, it can approximate any continuous spatio-temporal graph operator with an arbitrarily small $L^2$ error.

## A.3 GT-FNO Algorithm and Analysis of Computational Complexity

---

**Algorithm 1** Graph-Time Fourier Neural Operator (GT-FNO) Forward Pass

---

**Require:** Input graph signal $X \in \mathbb{R}^{B \times C_{\text{in}} \times N \times T}$
**Require:** Graph eigenvectors $\{\varphi_k\}$, Laplacian eigenvalues $\lambda$
**Require:** Absolute temporal embedding $h(t)$
**Require:** Model parameters: latent width $d$, spectral layers $L$, retained modes $K$
**Ensure:** Continuous reconstruction output $Y_{\text{CoSTL}} \in \mathbb{R}^{B \times C_{\text{out}} \times N \times T_{\text{out}}}$
1: $X_{\text{pe}} \leftarrow \text{SpaceTimePositionalEncoding}(X)$
2: $X_{\text{norm}} \leftarrow \text{LayerNorm}(X_{\text{pe}})$
3: $X_{\text{proj}} \leftarrow \text{Conv2D}(X_{\text{norm}}, d)$                      ▷ Lift to latent dimension $d$
4: $X_{\text{latent}} \leftarrow \text{SpectralConvT}(X_{\text{proj}}, \{\varphi_k\})$
5: $X_{\text{mlp}} \leftarrow \text{PointwiseFFN}(X_{\text{latent}})$
6: $X_{\text{lifted}} \leftarrow \sigma(X_{\text{proj}} + X_{\text{mlp}})$            ▷ $\sigma$ is a nonlinear activation
7: $X_{\text{gft}} \leftarrow \text{GFT}(X_{\text{lifted}}, \{\varphi_k\})$     ▷ Graph Fourier Transform to $(k, t)$ domain
8: $X_{\text{gtft}} \leftarrow \text{FFT}_{\text{time}}(X_{\text{gft}})$     ▷ 1D FFT to joint spectral domain $(k, \omega)$
9: $\tilde{h} \leftarrow \text{FFT}_{\text{time}}(h)$     ▷ Transform temporal embedding to frequency domain
10: **for** $l = 1$ **to** $L$ **do**
11:     $X_{\text{ssp}} \leftarrow \text{SpectralSegmentationPerception}(X_{\text{gtft}}, \lambda)$    ▷ Band-wise parameterization
12:     $g \leftarrow \sigma(W_g \tilde{h} + b_g)$                  ▷ Generate gating factor
13:     $X_{\text{tgssp}} \leftarrow g \odot X_{\text{ssp}}$             ▷ Time-gated modulation
14:     $X_{\text{gtft}} \leftarrow X_{\text{tgssp}}$         ▷ Update representations for the next layer
15: **end for**
16: $Y_{\text{time}} \leftarrow \text{IFFT}_{\text{time}}(X_{\text{tgssp}})$     ▷ Inverse temporal FFT back to $(k, t)$ domain
17: **if** $\{\varphi_k\}$ is complex **then**
18:     $Y_{\text{back}} \leftarrow \text{IGFT}(Y_{\text{time}}, \{\varphi_k\})$    ▷ Inverse GFT back to continuous spatiotemporal domain
19: **else**
20:     $Y_{\text{back}} \leftarrow \text{Real}(\text{IGFT}(Y_{\text{time}}, \{\varphi_k\}))$
21: **end if**
22: $Y_{\text{CoSTL}} \leftarrow \text{LinearProjection}(Y_{\text{back}}, C_{\text{out}}, T_{\text{out}})$
23: **return** $Y_{\text{CoSTL}}$

---

**Component Explanation:** The forward pass of GT-FNO first encodes the input spatio-temporal signal using a space-time positional encoding and normalizes it via layer normalization. A convolutional projection lifts the input to the desired latent width $d$, followed by a temporal spectral convolution (SpectralConvT) to capture initial correlations. A pointwise feed-forward network (FFN) refines the latent features, which are residually combined with the projected features through an activation function. The transformed features are then mapped to the spectral domain via a Graph Fourier Transform (GFT) using the graph eigenvectors $\Phi$, and further processed using a 1D FFT along the temporal dimension. Multiple spectral layers sequentially apply spatial spectral convolution (SpectralConvS) and a time-gating mechanism to model cross-scale dynamics. The output is mapped back to the spatio-temporal domain using inverse FFT and inverse GFT. Finally, a linear projection slices and maps the features to the required output steps to produce the final prediction.

**Computational and Model Complexity Analysis:** The GT-FNO framework involves several key components that contribute to its computational and parameter complexity. Let $B$ denote the batch size, $N$ the number of spatial nodes, $T$ the temporal length of input sequences, $C_{\text{in}}/C_{\text{out}}$ the input/output feature channels, $d$ the latent width, and $K$ the number of spectral modes retained in FFT operations.

1. *Positional Encoding and LayerNorm:* The space-time positional encoding and layer normalization require $\mathcal{O}(B \cdot C_{\text{in}} \cdot N \cdot T)$ operations and introduce negligible additional parameters compared to the main network.

2. *Convolutional Projection and Pointwise FFN:* The initial convolutional projection from $C_{\text{in}}$ to $d$ channels involves $\mathcal{O}(B \cdot d \cdot C_{\text{in}} \cdot N \cdot T)$ operations, with a parameter count of $d \cdot C_{\text{in}} \cdot k^2$ for kernel size $k$. The subsequent pointwise feed-forward network, applied per node and per time step, costs $\mathcal{O}(B \cdot N \cdot T \cdot d^2)$ operations and $2d^2$ parameters per FFN layer.

3. *Spectral Transformations:* Mapping features to the spectral domain using the GFT involves matrix multiplication with the graph eigenvector matrix of size $N \times N$, yielding $\mathcal{O}(B \cdot d \cdot N^2 \cdot T)$ operations. Temporal FFT/IFFT operations contribute $\mathcal{O}(B \cdot d \cdot N \cdot T \log T)$ complexity per layer, which is generally lower than the spatial spectral transform for large $N$.

4. *SpectralConvS Layers:* Each spectral convolution layer retains $K$ modes, incurring $\mathcal{O}(B \cdot d^2 \cdot K \cdot T)$ operations for linear transformations in the spectral domain, and $2d^2 \cdot K$ parameters per layer. Time-gating mechanisms introduce element-wise operations with $\mathcal{O}(B \cdot d \cdot N \cdot T)$ complexity and negligible parameters.

5. *Overall Complexity:* Combining all components, the total forward pass complexity is dominated by the spectral graph transform, yielding $\mathcal{O}(B \cdot d \cdot N^2 \cdot T + L \cdot B \cdot d^2 \cdot K \cdot T)$ operations, where $L$ is the number of spectral layers. The total parameter count is approximately $C_{\text{in}} \cdot d \cdot k^2 + L \cdot 2d^2 \cdot K + 2d^2$ per FFN, which grows linearly with the number of layers and quadratically with latent width $d$, but is fundamentally independent of $N$ due to mode truncation in the spectral layers.

**Efficiency Comparison.** By performing the graph Fourier transform via magnetic Laplacian eigendecomposition (complexity $\mathcal{O}(N^2 T)$) and a 1D FFT along the temporal axis (complexity $\mathcal{O}(NT \log T)$), we obtain a joint spatio-temporal spectrum. Applying the complex filter $\widetilde{Y}_k(\omega) = R_\phi(\lambda_k, \omega) \widetilde{X}_k(\omega)$ has a complexity of $\mathcal{O}(NTd^2)$, but our four-band parameter sharing significantly reduces the parameter count. Although this slightly exceeds the standard FNO's $\mathcal{O}(NTd)$ pointwise cost, it profoundly enhances spectral sensitivity and interpretability while remaining far below the PINN's $\mathcal{O}(Nd^3 + N^2)$ complexity. Thus, our approach extends neural operators to non-Euclidean graphs with only moderate computational overhead.

Furthermore, our proposed node–edge FiLM convolution (aggregation cost $\mathcal{O}(|E|d)$), combined with multiscale graph construction via Louvain clustering ($\mathcal{O}(N \log N)$) and hierarchical edge aggregation, effectively addresses discrete interaction modeling. This single-layer module extracts and fuses coarse, mid, and fine features at $\mathcal{O}(|E|d + Nd^2)$ (where $d^2 \ll N$). Compared to dynamic graph models with overall complexity $\mathcal{O}(N^2Td + NTd^2)$, our method significantly reduces the computational burden while effectively capturing long-range dependencies. In summary, the combination of GT-FNO and the adaptive multiscale interaction graph alleviates traditional bottlenecks, simultaneously enhancing both efficiency and generalization in spatio-temporal modeling.

## A.4 SPECTRAL ANALYSIS OF GRAPH-TIME DYNAMICS

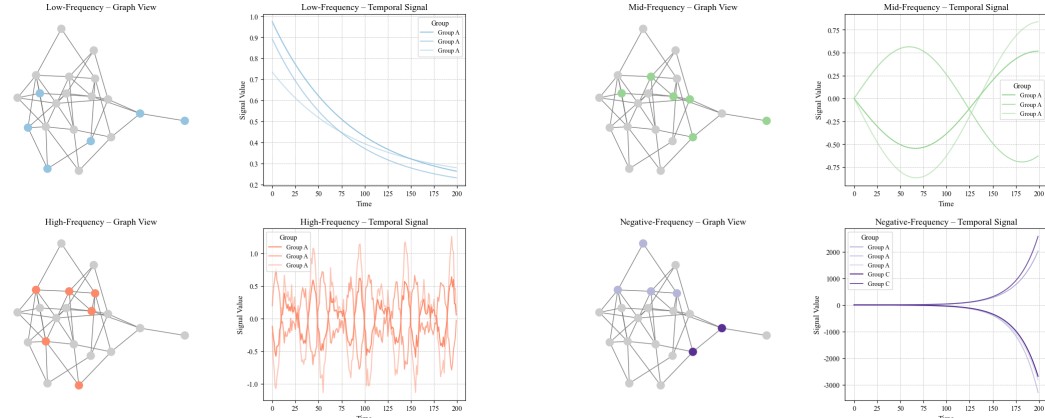

Figure 6: Visualization of graph-time dynamics across different spectral bands. The left panels (Graph View) display the spatial patterns of eigenvectors on the graph structure, while the right panels (Temporal Signal) illustrate their corresponding temporal evolution.

Consider an undirected graph with symmetric Laplacian $L = U\Lambda U^\top$, where $\Lambda = \text{diag}(\lambda_1, \ldots, \lambda_n)$ and $0 = \lambda_1 \leq \cdots \leq \lambda_n$. Let the graph-time linear system be

$$\partial_t x(t) = \Phi(t, L)x(t) + b(t),$$

where $x(t) \in \mathbb{R}^n$, and $\Phi(t, L)$ commutes with $L$ (e.g., a polynomial $\sum_{m=0}^{M} a_m(t)L^m$ or more generally a spectral function $\psi_t(L)$). In the graph Fourier domain $y(t) = U^\top x(t)$, the dynamics decouple completely:

$$\partial_t y_k(t) = \psi_t(\lambda_k)y_k(t) + \beta_k(t), \quad k = 1, \dots, n,$$

with $\beta_k(t) = u_k^\top b(t)$. For $b(t) = 0$, the solution is

$$y_k(t) = y_k(0)\exp\Big(\int_0^t \psi_s(\lambda_k)ds\Big),$$

so each graph frequency evolves independently, and its temporal behavior is fully determined by the corresponding eigenvalue $\lambda_k$ through $\psi_t(\lambda_k)$.

For typical diffusion/heat dynamics $\psi_t(\lambda) = -\kappa(t)g(\lambda)$ with monotone non-negative $g(\lambda)$, we have

$$|y_k(t)| = |y_k(0)|\exp\Big(-\int_0^t \kappa(s)g(\lambda_k)ds\Big).$$

Low-frequency modes (small $\lambda_k$) decay slowly, producing smooth, slowly varying graph signals. High-frequency modes (large $\lambda_k$) decay rapidly, and in the presence of observation noise or external input $b(t)$, manifest as fast, jittery variations. Intermediate frequencies decay at moderate rates but may exhibit oscillations in second-order or transport-type systems. For example, wave dynamics $\ddot{x} + c^2 Lx = 0$ decouple as $\ddot{y}_k + c^2 \lambda_k y_k = 0$ with solutions $y_k(t) = A_k \cos(c\sqrt{\lambda_k}t) + B_k \sin(c\sqrt{\lambda_k}t)$, showing oscillation frequencies increasing with $\sqrt{\lambda_k}$.

Discrete-time evolution $x_{t+1} = S_\tau(L)x_t$ with $S_\tau(L) = I - \tau G(L)$ gives spectral amplification factors $r_k = 1 - \tau G(\lambda_k)$, hence $y_k(t) = r_k^t y_k(0)$. Then: $0 < r_k < 1$ corresponds to monotone decay (stable low-frequency modes), $-1 < r_k < 0$ to alternating damped oscillations (mid-frequency modes), and $|r_k| > 1$ to divergence ("negative-frequency" behavior in the discrete-time signal). Note that $L$ remains positive semidefinite; negative signs originate from the discrete-time stepping operator or overly large explicit step sizes.

Intuitively, the graph Laplacian eigenvectors are ordered by the quadratic form $x^\top Lx$: low-frequency modes vary little across edges (smooth patterns), while high-frequency modes fluctuate rapidly along edges. Consequently, diffusion-like dynamics suppress high-frequency modes and preserve low-frequency structure, wave/transport dynamics map frequencies to faster oscillations, and discrete-time stepping may induce phase-flipped or unstable trajectories.

For operator learning, let the target evolution be $x(t) = T_t(L)x(0)$ with $T_t(L) = \exp\big(\int_0^t \psi_s(L)ds\big)$, and assume $\psi_s(\cdot)$ is continuous (or piecewise Lipschitz) on $[0, \lambda_{\max}]$. Partition $[0, \lambda_{\max}]$ into bands $\{I_b\}_b$ and approximate with piecewise linear spectral multipliers $\tilde{T}_{t,\theta}(\lambda)$ such that

$$\sup_{\lambda \in [0,\lambda_{\max}], t \in [0,T]} |T_t(\lambda) - \tilde{T}_{t,\theta}(\lambda)| \leq \delta.$$

Then for any input $X$,

$$\|T_t(L)X - \tilde{T}_{t,\theta}(L)X\|_2 \leq \delta\|X\|_2.$$

GT-FNO implements this idea with bandwise linear spectral operators and time gating, enabling controlled approximation of low-frequency decay, mid-frequency oscillations, high-frequency jitter, and discrete-time instability within bounded error.

In summary, each graph frequency determines the time scale and stability of its mode: decay rates scale with $\lambda$ under diffusion, oscillation frequencies scale with $\sqrt{\lambda}$ under wave dynamics, and discrete-time stability is governed by $|1 - \tau G(\lambda)| < 1$. Bandwise spectral operators provide a principled mechanism to manipulate these modes, yielding interpretable and accurate graph-time dynamic representations.

## A.5 DATASET DESCRIPTION

**Datasets.** We evaluate our model on three real-world datasets from traffic forecasting and air quality monitoring: **PEMS-BAY**, **SD**, and **KnowAir**. These datasets are widely used benchmarks in spatio-temporal learning, each introducing unique structural and temporal challenges. To ensure fairness

Table 5: Summary of datasets.

| Datasets | Range | Nodes | Time Interval | Length | Scene |
|----------|-------|-------|---------------|--------|-------|
| SD | San Diego | 716 | 5 mins | 35,040 | Traffic Prediction |
| PEMS-BAY | SF Bay Area | 325 | | 52,116 | |
| KnowAir | China | 184 | 3 hours | 11,667 | Air quality Forecasting |

and reproducibility, we follow the standard chronological segmentation protocol (Li et al., 2018), where the first 60% of the time series is used for training, the next 20% for validation, and the final 20% for testing, thus preventing temporal leakage.

**PEMS-BAY** is a benchmark dataset for traffic speed forecasting in the California Bay Area. It contains **325 loop detector sensors** deployed on major highways, recording average traffic speed every 5 minutes. The dataset spans from **January to June 2017**, covering more than **50,000 time steps** in total. This dataset is challenging due to the high variability in traffic conditions, including rush hours, holidays, and unexpected incidents. The sensor graph is constructed using road network distances, resulting in a weighted directed adjacency matrix where edge weights decay with geodesic distance.

**SD (San Diego)** is a subset of the LargeST benchmark (Liu et al., 2023), focusing on traffic speed prediction in the San Diego region. Unlike PEMS-BAY, SD incorporates a more **heterogeneous road network**, including both highways and local arterial roads, thus introducing a larger variation in spatial structures. The dataset consists of **716 sensors** sampled every 5 minutes, spanning **four consecutive months in 2021**. Its scale and diversity make it suitable for evaluating models' generalization capability across heterogeneous spatio-temporal patterns, particularly under non-stationary traffic dynamics.

**KnowAir** is a large-scale urban air quality forecasting dataset encompassing **184 main cities** across China. For each city, the dataset records the concentrations of multiple air pollutants (e.g., $PM_{2.5}$, $NO_2$, $SO_2$, $O_3$) alongside meteorological factors such as wind speed, temperature, and humidity. The observations are sampled every **3 hours** and span a four-year period from **2015 to 2018**, yielding over **11,600 time steps** per city. Compared with traffic data, air quality signals evolve on slower temporal scales but are heavily influenced by complex physical processes (e.g., wind-driven diffusion, chemical transformation). This dataset is particularly challenging for graph-based models because the spatial correlations are highly dynamic and strongly influenced by external meteorological variables.

**Preprocessing.** All datasets are normalized using z-score normalization based on training set statistics. Missing values, which occur sporadically in sensor readings, are imputed using temporal nearest-neighbor interpolation. The input horizon is fixed at **12 time steps** (corresponding to 1 hour for PEMS-BAY/SD and 12 hours for KnowAir), and the prediction horizon is set to **12 time steps**, consistent with prior work. The final data tensors are formatted as $X \in \mathbb{R}^{B \times N \times T \times F}$, where $B$ is the batch size, $N$ the number of sensors, $T$ the temporal window length, and $F$ the number of features per node.

### A.6 EVALUATION METRICS

To rigorously evaluate the performance of the proposed model on both interpolation and extrapolation tasks, we adopt several task-specific regression metrics that comprehensively reflect the model's ability to capture the underlying dynamics of irregular time series. The evaluation metrics are detailed as follows:

**Mean Absolute Error (MAE).** MAE measures the average magnitude of the absolute differences between predicted values and ground truth values, defined as:

$$\text{MAE} = \frac{1}{NT} \sum_{i=1}^{N} \sum_{t=1}^{T} |\hat{y}_{i,t} - y_{i,t}|,$$

where $N$ and $T$ denote the number of sensors and time steps, respectively; $\hat{y}_{i,t}$ is the predicted value, and $y_{i,t}$ is the ground truth. MAE is robust to outliers and reflects the average absolute deviation.

**Root Mean Squared Error (RMSE).** RMSE quantifies the square root of the average squared differences between predicted and actual values, placing greater emphasis on larger errors:

$$\text{RMSE} = \sqrt{\frac{1}{NT} \sum_{i=1}^{N} \sum_{t=1}^{T} (\hat{y}_{i,t} - y_{i,t})^2}.$$

RMSE penalizes large deviations more than MAE, making it suitable for scenarios where high-magnitude errors are critical.

**Mean Absolute Percentage Error (MAPE).** MAPE measures the average absolute percentage difference between predictions and actual values:

$$\text{MAPE} = \frac{100\%}{NT} \sum_{i=1}^{N} \sum_{t=1}^{T} \left| \frac{\hat{y}_{i,t} - y_{i,t}}{y_{i,t}} \right|,$$

where the division is element-wise. MAPE provides an interpretable, percentage-based error metric but may become unstable when the ground truth values approach zero.

These metrics provide complementary perspectives on forecasting accuracy and robustness under various conditions.

### A.7 HYPERPARAMETER SENSITIVITY ANALYSIS

To assess the robustness and adaptability of the proposed CoSTL and AMI modules, we perform a sensitivity analysis on four key hyperparameters across the KnowAir and PEMS-BAY datasets: the GT-FNO hidden dimension (`gfno_hidden`), the spectral segmentation threshold (`spectral_segment`), the number of global edges per node (`topk_edge`), and the number of edge-node coupling convolution layers in the multi-scale dynamic graph convolution network(MSGCN), denoted as the ECC layer depth (`ecc_layer`).

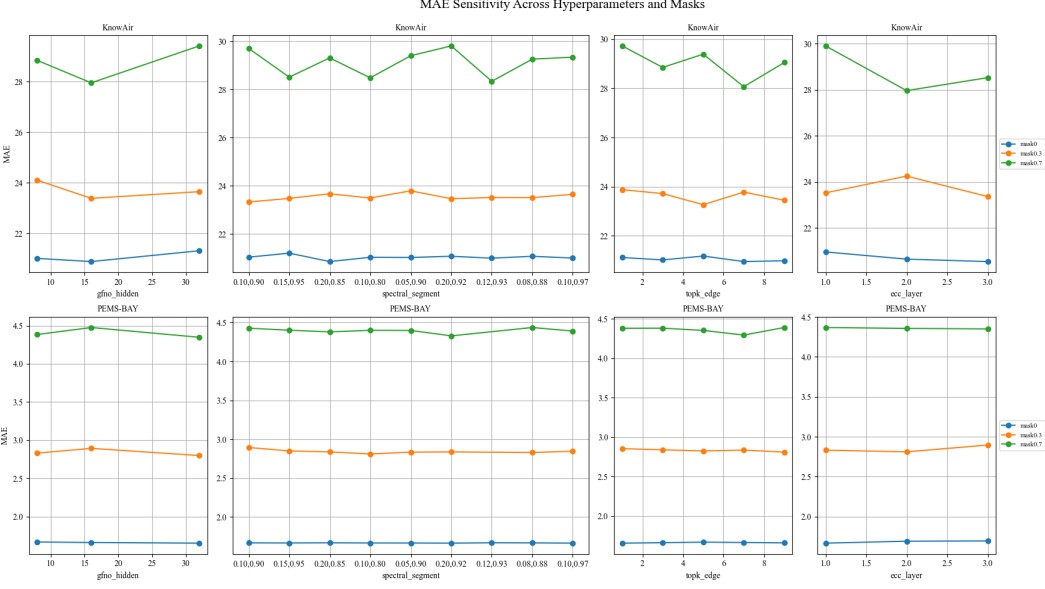

Figure 7: Hyperparameter sensitivity analysis of PhySTA on the PEMS-BAY and KnowAir datasets.

As shown in Figure 7, larger `gfno_hidden` values improve MAE on PEMS-BAY, while KnowAir exhibits greater variance, possibly due to overfitting or weaker spatial correlations. The spectral segmentation parameter (`spectral_segment`) shows stable performance across a wide range of

threshold values, demonstrating AMI's robustness to segmentation granularity, especially on PEMS-BAY. Increasing the number of global edges per node (`topk_edge`) slightly degrades performance under high masking, which may result from noise amplification in over-connected structures. Finally, stacking more graph convolution layers (i.e., a larger `ecc_layer`) tends to increase MAE on KnowAir, likely due to over-smoothing or gradient issues, whereas PEMS-BAY benefits moderately from increased depth.

These results confirm PhySTA's overall stability, while highlighting that architectural depth and global connectivity should be tuned carefully under sparse or irregular input conditions.

**Scalability Analysis.** The scalability of **PhySTA** is validated across datasets with diverse spatial scales, temporal resolutions, and domains, including traffic forecasting (PEMS-BAY, SD) and air quality prediction (KnowAir), as summarized in Table 5. PhySTA consistently performs well across small (184 nodes in KnowAir), medium (325 nodes in PEMS-BAY), and large-scale graphs (716 nodes in SD), demonstrating its adaptability to increasing graph sizes and data volumes.

As shown in Table 1, PhySTA also maintains strong predictive performance under varying levels of node masking, simulating partial observability. On SD, it achieves the lowest MAE of **20.64** with full data and a competitive MAE of **96.09** at 70% masking. On KnowAir, PhySTA reaches MAEs of **22.89** and **27.19** at 30% and 70% masking, respectively, and similarly outperforms baselines on PEMS-BAY with scores of **2.75** and **4.25**. These results show that PhySTA scales effectively with spatial and temporal complexity while remaining robust under data sparsity, confirming its suitability for real-world deployment in large and partially observable environments.

# B USE OF LARGE LANGUAGE MODELS (LLMS)

In the preparation of this work, we utilized large language models (LLMs) primarily as general-purpose assistive tools for English language polishing, grammar and formatting checks, and for guidance in reviewing background knowledge related to operator learning. The LLM did not contribute to the core research ideas, algorithm design, theoretical derivations, or experimental analyses. All scientific content, claims, and results presented in this paper were generated and verified by the authors. We take full responsibility for the accuracy, originality, and integrity of the content, including any portions refined with the assistance of LLMs. No LLM was listed as an author.

