# OpenReview forum: "Enabling arbitrary inference in spatio-temporal dynamic systems: A physics-inspired perspective"
_ICLR.cc/2026/Conference — ICLR 2026 Poster_

### Official Review · Reviewer_ucx5 · 2025-10-27

**Soundness:** 3
**Presentation:** 2
**Contribution:** 3
**Rating:** 6
**Confidence:** 4

**Summary:**

This paper proposes PhySTA (Physics-inspired Spatio-Temporal Learning for Arbitrary Inference), a framework that models continuous spatio-temporal dynamics on graph-structured data by integrating operator learning with multi-scale graph neural networks. The core idea is to bridge the gap between continuous real-world dynamics and discrete sensor observations, enabling reliable inference in unobserved regions.

The framework comprises two main modules:
Continuous Operator-based Spectrum-Temporal Learning (CoSTL): Uses a Graph-Time Fourier Neural Operator (GT-FNO) and Time-Gated Spectral Segmentation Perception to model continuous dynamics in the joint spectral domain

Adaptive Multi-scale Interaction (AMI): Employs a novel Node-Edge Coupled Convolution and Multi-scale Subgraph Partition (coarse-mid-fine hierarchy) to capture discrete multi-scale interactions and refine the continuous predictions.

**Strengths:**

Strong Theoretical and Architectural Novelty: The approach successfully extends the Fourier Neural Operator (FNO), a tool for continuous Euclidean domains, to non-Euclidean graph domains (GT-FNO) using the magnetic Laplacian and a joint graph-time spectral decomposition. This is a significant theoretical advance in spatio-temporal GNNs.

Robustness to Data Sparsity: PhySTA consistently achieves top performance across all datasets under varying degrees of node masking (up to $70\%$ sparsity). This directly validates its core claim of enabling robust "arbitrary inference" in unobserved regions

fficiency and Low Overhead: PhySTA achieves state-of-the-art accuracy while using a lower number of parameters ($123,474$) and less GPU memory ($6,042 \text{MB}$) compared to most deep GNN baselines like AGCRN and STG-ODE

**Weaknesses:**

1. Scalability Bottleneck of Spectral Decomposition: The paper acknowledges that the Graph Fourier Transform (GFT) step, which involves spectral decomposition (magnetic Laplacian), can be computationally demanding on very large graphs9. Since GFT's complexity is $O(N^2)$, this $N^2$ dependency limits the framework's scalability for massive real-world graphs (e.g., city-scale mobility or power grids with $N \gg 1000$). It is crucial to discuss concrete techniques like Nyström approximation or sparse spectral methods to mitigate the $O(N^2)$ bottleneck

2. Ambiguity of Multi-scale Aggregation: The AMI module's multi-scale single-layer modeling (coarse, mid, fine graphs) claims to capture complex interactions efficiently. However, the Louvain community detection used for subgraph creation is non-differentiable, potentially complicating end-to-end training. The authors should clarify how the node2center mapping and the three-level graph structure are integrated into the differentiable training pipeline.

**Questions:**

Addressing GFT Scalability (The Major Bottleneck): The paper acknowledges that the $O(N^2)$ complexity of the full Graph Fourier Transform (GFT) limits scalability. Since current spatio-temporal benchmarks scale up to 716 nodes (SD), the $N^2$ barrier remains a practical constraint for city-scale graphs ($N>5000$). Could the authors propose and experimentally validate a concrete, scalable approximation technique within the GT-FNO architecture (e.g., using a localized sparse spectral filter, Nyström approximation on the magnetic Laplacian, or spectral compression) to push the scalability beyond the current constraint?


Differentiability and Justification of Multi-scale Graph Construction: The Adaptive Multi-scale Interaction (AMI) relies on Louvain community detection to generate coarse and fine subgraphs. Since community detection is generally non-differentiable, how is the graph hierarchy generation process integrated into the end-to-end training pipeline? Furthermore, given the complexity of AMI, can the authors provide a side-by-side comparison (e.g., in the Appendix) of the performance gains versus a simpler, fully differentiable multi-scale strategy, such as one based on standard differentiable pooling (DiffPool/TopK)?


Detailed Analysis of Spectral Component Contributions (TGSSP): The ablation study suggests that the specialized Time-Gated Spectral Segmentation Perception (TGSSP) module contributes only minor gains compared to other components. Given the module's complexity (bandwise parameterization, time-gating, complex eigenvalues/magnetic Laplacian), can the authors provide a more detailed visualization or analysis of what TGSSP learns? For example, showing how the learned per-mode scaling factor ($\alpha_k$) and the time-gating factor ($g(\omega)$) prioritize or suppress specific frequency bands over time would better justify the module's role in modeling non-stationary dynamics.

More applications: could it be applied to weather and climate forecasting?

---

> ### Author Response · Authors · 2025-11-21
> **Part 1 of the Response (1/2)**
>
> Dear Reviewer ucx5
>
> We sincerely thank you for the constructive comments. Below we address the concerns regarding the scalability of GFT, the differentiability of the AMI module, the interpretability of TGSSP, and the framework's broader applicability.
>
> ### 1. Scalability of GFT and spectral operations
> We have introduced the recommended Nyström-based approximation to analyze and refine our solution.
>
> 1.  **Algorithmic implementation and empirical comparison:** We implemented a Nyström approximation by sampling anchor nodes ($S \\ll N$), which achieves a linear complexity of $O(S^2 N + S^3)$ and decouples the parameter count from the total node number. We compared this approximation against the full GT-FNO on the PEMS-BAY dataset. The results show that while the approximation effectively reduced the parameter count ($\sim$86K vs. $\sim$123K), it resulted in a performance degradation, with the Test MAE increasing from **1.66(Original)** to **1.96 (Nyström)**.
> 2.  **Analysis of the scalability-precision trade-off:** The observed performance drop highlights a critical trade-off. The Nyström method predominantly captures low-frequency principal components, inherently filtering out high-frequency details. This renders our proposed `energy_splits` strategy (which explicitly models Low/Mid/High bands) ineffective, as the specific high-frequency information required for independent band processing is lost during approximation. Thus, while approximation methods enable massive scaling, they sacrifice the precise high-frequency modeling capabilities that drive our model's superior accuracy.
>
> Thus, while Nyström approximations offer a viable path for massive scaling, the full GFT remains superior for tasks where precise high-frequency modeling is critical to accuracy.
>
> ### 2. Differentiability and design rationale of the AMI module
> The non-differentiability of the Louvain clustering algorithm does not impede end-to-end training because graph partitioning is executed as a **one-time offline preprocessing step**.
>
> **Gradient flow:** Since clustering is performed before training begins and the resulting subgraphs are fixed, the adjacency matrices serve as constant inputs. Consequently, gradients do not need to flow through the clustering algorithm itself, ensuring the differentiability of the subsequent learning process.
>
> We prioritized this static approximation over learnable pooling (e.g., DiffPool or TopK) based on two key factors. Urban spatial structures (functional zones) evolve very slowly compared to high-frequency traffic dynamics. While we use static partitioning for the stable macro-structure, the rapid dynamic changes are effectively handled by our **TGSSP** and **feature-driven interaction** modules. Louvain clustering provides distinct, physically interpretable community structures without the instability or memory overhead associated with learning dense assignment matrices in differentiable pooling methods.
>
> However, we appreciate your suggestion and acknowledge that in small-graph scenarios, employing learnable parameters to control subgraph generation could theoretically achieve superior predictive performance, albeit with an acceptable trade-off in computational efficiency.

---

> ### Author Response · Authors · 2025-11-21
> **Part 2 of the Response (2/2)**
>
> ### 3. Contribution and interpretability of TGSSP
> To address the reviewer’s concern about the contribution and interpretability of TGSSP, we added a dedicated case study section titled **“Spectral Specialization and Temporal Adaptation” (Section 4.4)**. There, we analyze its two key parameter groups—mode-wise scaling factors $(\alpha_k)$ and temporal gating activations—and how they jointly determine spectral–temporal behavior during training and inference.
>
> 1. **Spectral specialization.** By tracking $\alpha_k$ throughout training, we observe a progressive “spectral selection” effect: informative high-band modes are consistently amplified, while less relevant modes are gradually suppressed. This shows that TGSSP learns to specialize in a subset of task-relevant spectral components rather than applying uniform weighting across frequencies.
>
> 2. **Frequency-adaptive temporal gating.** Evaluation-time visualizations reveal a stabilized gating strategy. Low-frequency components follow an almost uniform pass-through policy that preserves global trends, whereas high-frequency components exhibit high-variance, selective gating tuned to local irregularities. Gate activations spike in sync with transient bursts of high-frequency energy and decay as the signal stabilizes, effectively acting as a spectral-domain attention mechanism that enhances signal–noise contrast.
>
> 3. **Band-specific temporal focus.** We further examine the learned temporal kernels and find clearly differentiated temporal receptive fields across frequency bands. Low-frequency bands concentrate on longer-term contexts, while higher bands focus on shorter, more transient patterns. This confirms that TGSSP learns band-specific temporal focus, which is crucial for modeling multi-scale, non-stationary dynamics.
>
> These observations provide a concrete, mechanism-level explanation of how TGSSP coordinates spectral specialization with temporal adaptation, clarifying both its functional role and its contribution to the overall performance of the model.
>
> ---
>
> ### 4. Applicability to broader domains
> The framework has been validated on environmental monitoring tasks, though distinct physical characteristics of global meteorological grids require separate future investigation.
> We have verified the model's cross-domain capabilities using the **KnowAir [8]** dataset, which records PM$_{2.5}$ concentrations from distributed air-quality monitoring stations together with meteorological covariates (e.g., wind speed, temperature, humidity) that are correlated with the prediction target. Our model is explicitly designed to accommodate such multi-variate drivers, confirming its effectiveness in modeling diffusion-based environmental processes beyond traffic flow.
>
> Regarding large-scale meteorological datasets such as **ERA5 [9]** and **WeatherBench [10]**, these are defined on massive latitude-longitude grids with more explicit physical dependencies. While our model is theoretically compatible, the specific structural properties of these global grids warrant a dedicated study.
> We therefore establish cross-domain viability via KnowAir in this work and plan to address global climate modeling in future extensions.
>
> ---
> Yours sincerely,
>
> The authors of Paper 18391
>
>
>
> ### References
> **[8] KnowAir:** Wang, Shuo, et al. "Pm2. 5-gnn: A domain knowledge enhanced graph neural network for pm2. 5 forecasting." *Proceedings of the 28th international conference on advances in geographic information systems*. 2020.
>
> **[9] ERA5:** Hersbach, Hans, et al. "The ERA5 global reanalysis." *Quarterly journal of the royal meteorological society* 146.730 (2020): 1999-2049.
>
> **[10] WeatherBench:** Rasp, Stephan, et al. "WeatherBench: a benchmark data set for data‐driven weather forecasting." *Journal of Advances in Modeling Earth Systems* 12.11 (2020): e2020MS002203.

---

> ### Author Response · Authors · 2025-11-26
> **Thanks for your valuable suggestions and Looking forward to further discussion**
>
> Dear Reviewer ucx5,
>
> We sincerely thank you for the constructive comments. Below we address your concerns regarding the scalability of GFT, the differentiability of the AMI module, the interpretability of TGSSP, and the framework’s broader applicability.
>
> **Part 1 of the Response (1/2):** We analyze the scalability of GFT and related spectral operations by implementing the suggested Nyström approximation, comparing its complexity and accuracy against the full GT-FNO on PEMS-BAY, and clarifying the resulting scalability–precision trade-off. We also explain why the full GFT is retained for tasks that require precise high-frequency modeling. In addition, we clarify that the AMI module remains fully differentiable because Louvain clustering is performed only once as offline preprocessing, and we motivate this design choice over learnable pooling in terms of stability, interpretability, and computational efficiency.
>
> **Part 2 of the Response (2/2):** We improve the interpretability of TGSSP through a new case-study section on “Spectral Specialization and Temporal Adaptation,” illustrating how mode-wise scaling and temporal gating jointly produce spectral selection, frequency-adaptive gating, and band-specific temporal focus. We also discuss applicability beyond traffic flow by validating the framework on the KnowAir air-quality dataset with rich meteorological covariates and by outlining how it can be extended to large-scale meteorological grids such as ERA5 and WeatherBench in future work.
>
> Thank you again for your time, constructive suggestions, and thoughtful evaluation. As the discussion phase is progressing, we would be truly grateful if you could kindly take a moment to engage further when convenient.
>
> ---
>
> Yours sincerely,
>
> The authors of Paper 18391

---

### Official Review · Reviewer_5Bv2 · 2025-10-27

**Soundness:** 3
**Presentation:** 3
**Contribution:** 3
**Rating:** 4
**Confidence:** 3

**Summary:**

This paper focuses on addressing the challenge of arbitrary inference in spatiotemporal dynamic systems, where existing methods either fail to generalize to unseen regions and complex graph structures or are confined to Euclidean grids. The proposed framework PhySTA integrates two innovative core modules: Continuous Operator-based Spectrum-Temporal Learning (CoSTL), which extends neural operators to non-Euclidean domains via a Graph-Time Fourier Neural Operator (GT-FNO) and Time-Gated Spectral Segmentation Perception for continuous dynamics modeling, and Adaptive Multi-scale Interaction (AMI) that constructs multiscale subgraphs and uses node-edge coupled convolution to capture discrete interactions. Theoretically, GT-FNO is proven to have universal approximation capability for continuous spatio-temporal graph operators with controllable \(L^2\) error. Experimentally, on traffic and air quality benchmarks (PEMS-BAY, SD, KnowAir), PhySTA achieves state-of-the-art accuracy across various missing data ratios, reduces computation cost significantly, and has fewer parameters and lower GPU memory consumption compared to baselines, demonstrating robust generalization for arbitrary inference even in sparse sensor scenarios.

**Strengths:**

1. The paper is well-structured, and its notations are generally clear.
2. The proposed approach demonstrates performance improvements in most experimental settings compared to recent baseline methods.

**Weaknesses:**

1. In the motivation section, the authors propose that the modeling of continuous changes relies on graph spectral modeling, but the correlation between the two is not clearly established.
2. It is not specified where the truncation operation is performed.
3. In Equation 7, the graph is divided into three layers: coarse, mid, and fine. However, these subgraphs only exist on partial nodes. Therefore, it remains unclear whether the extracted features (X_coarse, X_mid, X_fine) have missing node features respectively.
4. Issues with notations and formatting:
   - In Line 256, there is a missing space before "Inspired"; in Line 264, there is a missing space before "The".
   - In Equation 8, how are the two predicted values (y_costl and y_ami) obtained? Is the result of Equation 5 equivalent to Y_AMI?
5. How do continuous spectrum-temporal modeling and adaptive multi-scale interaction handle dynamic topology changes respectively? At present, the study seems to treat the underlying graph (i.e., adjacency matrix A) as static, without corresponding discussions on dynamic scenarios.

**Questions:**

see details in weakness

---

> ### Author Response · Authors · 2025-11-21
> **Part 1 of the Response (1/2)**
>
> Dear Reviewer uXuB
>
>
> We sincerely appreciate your insightful and constructive comments. We have carefully revised the manuscript and clarified key concepts, theoretical connections, modeling details, and formatting issues. Below we provide concise, structured responses to each concern.
>
> ### 1. Continuous operator learning and spectral–temporal modeling
>
> Continuous operator learning can be viewed as learning a kernel over function spaces, and our spectral framework is designed exactly for this purpose.
> 1. **Neural operators.** Traditional deep models learn finite-dimensional vector-to-vector mappings, even though these vectors are merely discrete samples of underlying continuous functions. As discussed in **Appendix A.2**, neural operators instead aim to approximate solution operators between function spaces, i.e., an “infinite-dimensional” mapping. In this view, learning the operator is mathematically equivalent to learning its associated integral kernel.
>
> 2. **Spectral parameterization** Our graph spectral modeling provides an orthogonal, compact parameterization of this kernel directly in the graph spectral domain, as detailed in the “Continuous Reconstruction” paragraph of **Section 3.3**.
>
> Beyond enabling continuous operator learning, the proposed **Time-Gated Spectral Segmentation Perception**(TGSSP) module further leverages the link between spectral bands and system dynamics: by partitioning the spectrum into learnable bands, it assigns different frequency ranges to distinct dynamical patterns. A concise visualization study (**Section 4.4**) shows that TGSSP selectively amplifies informative modes and applies frequency-dependent temporal gating, yielding interpretable spectral specialization and temporal adaptation that align with its observed performance gains.
>
> ---
> ### 2. Clarification of the spectral truncation mechanism
> We have revised the manuscript to provide a precise description of the spectral truncation and segmentation mechanism presented in **Section 3.3 (Time-Gated Spectral Segmentation Perception)**.
>
> We clarify that truncation is performed strictly *after* the transformation into the joint spectral domain. Both the Graph Fourier Transform (GFT) and temporal Fast Fourier Transform (FFT) are executed on the input data first. The resulting joint spectral coefficients are then partitioned, processed, and fused.
>
> As detailed in **Appendix A.4**, different graph spectral modes correspond to distinct physical dynamical components. Guided by this analysis, we segment the spectrum into negative, low, mid, and high-frequency bands. This allows us to apply  independent weights for dominant low frequencies to capture trends, and shared kernels with scaling factors for high frequencies to capture transients.
>
> The specific thresholds for these bands are not arbitrary; they are constructed based on information-theoretic criteria (analyzing spectral energy entropy) and were finalized through experimental validation to ensure optimal separation of dynamical patterns.
>
>
> ---
>
> ### 3. Feature completeness under multi-scale graph partitioning
> We clarify that missing node features are masked at the input stage and the holistic integration of multi-scale graphs ensures that no information is lost. Detailed mechanisms are provided in **Section 3.4 (Multi-scale Single-Layer Long-Range Modeling)**.
>
> 1.  **Inference mechanism for unseen nodes:** In the unseen-node inference task, target nodes are masked (features set to zero) at the input stage but remain topologically present in the graph structure. The subgraphs infer the states of these zero-padded nodes not merely through simple propagation, but by leveraging patterns learned from observed nodes via designed interaction relationships. During the testing phase, we mask the features of these nodes while retaining their labels strictly for evaluation purposes.
> 2.  **Multi-scale information preservation:** While coarse-scale graphs (induced by clustering) abstract away local details to model global structural interactions, this is fully compensated by the mid- and fine-scale layers which preserve original connectivity and long-range interactions. Consequently, this complementary multi-scale design ensures that, in aggregate, no critical feature information is lost, while the recovery of missing values actively encourages the generative capacity of the learning framework.

---

> ### Author Response · Authors · 2025-11-21
> **Part 2 of the Response (2/2)**
>
> ### 4. Formatting and equation corrections
> Formatting inconsistencies and equation definitions have been fully corrected to ensure clarity and precision throughout the manuscript.
>
> 1.  **Reorganized technical narrative:** We restructured the method section to ensure a smoother logical flow. To reduce ambiguity, abstract symbols have been replaced with operation-specific notation. For instance, intermediate features previously denoted generically as $\\hat{X}$ and $\\tilde{X}$ are now explicitly labeled as $X_{\\text{gft}}$ and $X_{\\text{gtft}}$, directly reflecting their derivation from the Graph Fourier Transform and Joint Fourier Transform stages.
> 2.  **Standardized mathematical notation:** We enforced strict consistency in symbol usage throughout the paper: Input tensors are now uniformly denoted as $X_{1:T}$ (explicitly declaring time dimensions) rather than the generic $X$. Spectral segmentation indices have been unified under the notation $\mathcal{I}{\cdot}$ (e.g., $\mathcal{I}{\text{low}}$, $\mathcal{I}{\text{mid}}$, $\mathcal{I}{\text{high}}$) across all definitions. Spacing and operator typesetting have been standardized in all equations.
> 3.  **Redrawn key figures:** Figures 2 and 3 have been revised to enhance stand-alone interpretability. We added explicit labels for intermediate variables (matching the updated notation like $X_{\\text{gft}}$) at the connection points between modules, allowing readers to visually trace the mathematical operations within the architecture.
> 4.  **Clarification of equation (11):** Regarding the previously ambiguous Equation (8) (now renumbered as Equation 11), we have clarified that $y_{\\text{costl}}$ and $y_{\\text{AMI}}$ represent the distinct outputs of the Graph Fourier Operator and the Adaptive Multi-scale Interaction module, respectively.
> 5.  **Explicit formulation of AMI:** To ensure the computational workflow is transparent, the updated manuscript now explicitly states the construction of the interaction output:
>     $$
>     Y_{\\text{AMI}} = \\text{Concat}(X_{1:T}, X_{\\text{coarse}}, X_{\\text{mid}}, X_{\\text{fine}}) \\in \\mathbb{R}^{N \\times (C + 3H)}
>     $$
>     This definition mathematically confirms how multi-scale features are aggregated before final fusion.
>
> These revisions substantially improve internal consistency and make the computational workflow easier to interpret.
>
> ---
>
> ### 5. Static topology with dynamic structural evolution
> While we utilize a static adjacency matrix $A$ as the physical skeleton, our model effectively captures dynamic structural evolution by modulating signal propagation through **time-gated spectral filtering** and **feature-adaptive interactions**.
>
> 1.  **Continuous spectral temporal modeling:** Although the graph spectral basis (derived from static $A$) is fixed, our **Time-Gated Spectral Segmentation Perception (TGSSP)** module dynamically modulates the energy distribution across frequency bands via temporal gating. This allows the model to represent evolving dynamical patterns and non-stationary shifts without requiring the explicit re-computation of the Laplacian matrix at every step.
>
> 2.  **Adaptive multi-scale interaction:** The multi-scale interaction module transcends the static topology by incorporating **feature-driven dynamics**. Through node-edge coupling, the effective connectivity strength is conditioned on real-time node features rather than fixed edge weights. This injects dynamic adaptability into the interaction mechanism, simulating the behavior of a time-evolving graph.
>
> While these implicit mechanisms allow our model to handle non-stationary spatiotemporal patterns on a static physical backbone, we greatly appreciate this constructive suggestion. Motivated by your comments, we have surveyed recent work on dynamic graph learning and operator-based models, which has provided us with valuable insights and concrete ideas (e.g., learnable time-varying adjacency matrices and jointly evolving operators). We plan to incorporate such explicit dynamic graph learning schemes in future extensions to more thoroughly capture topological evolution.
>
> ---
> Yours sincerely,
>
> The authors of Paper 18391

---

> ### Author Response · Authors · 2025-11-26
> **Thanks for your valuable suggestions and Looking forward to further discussion**
>
> Dear Reviewer 5Bv2,
>
> We sincerely appreciate the thoughtful and detailed review you provided for our manuscript. Following your insightful comments, we have carefully prepared a point-by-point response addressing your concerns:
>
> **Part 1 of the Response (1/2):** We clarify the theoretical foundations of our approach by strengthening the connection between continuous operator learning and spectral–temporal modeling, and by providing a precise description of the spectral truncation and segmentation mechanism in TGSSP. We also explain how multi-scale graph partitioning preserves feature completeness and supports unseen-node inference through holistic integration of coarse, mid, and fine scales.
>
> **Part 1 of the Response (2/2):** We correct formatting inconsistencies and equation definitions, reorganize the technical narrative with standardized notation and redrawn figures, and explicitly formulate the adaptive multi-scale interaction module. In addition, we discuss how time-gated spectral filtering and feature-adaptive interactions enable dynamic structural evolution on a static topology, and outline future directions toward explicit dynamic graph learning.
>
> Thank you again for your time, constructive suggestions, and thoughtful evaluation. As the discussion phase is progressing, we would be truly grateful if you could kindly take a moment to engage further when convenient.
>
> ---
> Yours sincerely,
>
> The authors of Paper 18391

---

### Official Review · Reviewer_sBzk · 2025-10-30

**Soundness:** 3
**Presentation:** 3
**Contribution:** 2
**Rating:** 6
**Confidence:** 3

**Summary:**

The paper proposes PhySTA, a physics-inspired framework that unifies continuous operator learning with graph-based spatio-temporal modeling. It introduces two main modules:
(1) a Graph–Time Fourier Neural Operator (GT-FNO) equipped with Time-Gated Spectral Segmentation Perception (TGSSP) for modeling continuous spectral dynamics on graphs, and
(2) an Adaptive Multi-Scale Interaction (AMI) mechanism that captures multi-scale node–edge relationships via coupled convolution and hierarchical graph construction.
A Continuity–Discreteness Interaction Module (CDIM) further fuses both continuous and discrete predictions for arbitrary inference in unobserved regions.
Experiments on large-scale traffic and air-quality datasets demonstrate strong accuracy, robustness, and efficiency compared with several state-of-the-art baselines.

**Strengths:**

1. Novel integration of physics-inspired operator learning and GNNs:
The proposed GT-FNO extends Fourier Neural Operators to non-Euclidean graphs, enabling continuous modeling over directed graphs—a clear conceptual innovation.

2. Multi-scale and coupled graph design:
The AMI module effectively captures long-range, multi-level dependencies within a single layer, addressing over-smoothing and inefficiency issues seen in deep GNNs.

3. Strong empirical performance and efficiency:
PhySTA achieves consistent improvements across datasets with fewer parameters and memory cost (≈74% FLOP reduction), showing excellent trade-offs between accuracy and scalability.

4. Comprehensive experiments and ablation analysis:
The inclusion of multiple datasets, mask ratios, and detailed component ablations provides good evidence of robustness and interpretability.

**Weaknesses:**

* Limited comparison to recent operator-based or physics-informed baselines:
The paper mainly compares against classical and GNN-based methods (STGCN, DGCRN, etc.), but omits recent neural operator or PDE-based baselines such as Graph Neural Operator (Li et al., 2023) or Geo-FNO (Li et al., 2024). These would strengthen the claim of operator generalization.

* Writing quality and presentation:
The exposition is heavy and sometimes unclear, especially in the methodology section. Some mathematical notations are inconsistent, and figures (e.g., Fig. 2, Fig. 3) are not fully self-explanatory. The authors could simplify and streamline the presentation for readability.

* Ablation and interpretability could be expanded:
Although the ablation table is informative, qualitative insights on how each frequency band or subgraph level contributes to the final prediction are missing. Visualizations of spectral energy distribution or temporal gating behavior would enhance interpretability.

* Scalability limitation not sufficiently addressed:
The reliance on magnetic Laplacian spectral decomposition may hinder scalability for very large graphs. While this is briefly mentioned in the limitations, empirical evaluation on larger graphs would make the claim more convincing.

* Incomplete baseline coverage:
Some recent transformer-based and neural-operator hybrid methods (e.g., Graphormer, SpaceTimeFormer) are missing from comparison, which may weaken the “state-of-the-art” claim.

**Questions:**

See weaknesses.

---

> ### Author Response · Authors · 2025-11-21
> **Part 1 of the Response (1/2)**
>
> Dear Reviewer sBzk
>
> We sincerely appreciate your insightful comments, which have greatly helped us refine the manuscript. Below we provide detailed responses to each concern. In summary, we enriched the baseline coverage, improved clarity through cleaner notation and revised figures, added new analyses to better interpret TGSSP, clarified scalability on large graphs, strengthened the link between continuous operators and graph spectra, and provided a clearer explanation of the truncation mechanism. These updates substantially enhance completeness, readability, and rigor.
>
> ### 1. Expanded baseline comparison
> We have expanded and clarified the scope of related work to ensure complete baseline coverage.
> We have expanded and clarified the scope of related work to ensure complete baseline coverage.
> 1. **Operator-based baselines:** We adapt **GNO [1]** and **Geo-FNO [2]** to graph domains by replacing the GTFNO operator with their formulations. On the air quality dataset **KnowAir**, we evaluate them both as standalone predictors and as substitutes for our operator. In all cases, our full model achieves lower MAE, MAPE, and RMSE, and the performance drop under operator replacement highlights the additional gain from the adaptive interaction module.
>
> 2. **Transformer and DE-based baselines:** Beyond the **STGODE [3]** baseline (Table 1 on page 8), we reproduce **Graphormer [4]** and the PDE-based **STDen [5]**, and report their forecasting and unseen-node inference results on the **PEMS-BAY** traffic dataset. Across all mask ratios, these attention-only and PDE-based models are consistently worse than **PhySTA**(our model), confirming the benefit of coupling a neural operator with a graph network rather than relying on either component alone.
>
> **Table 1: Forecasting performance comparison under different Mask Ratios (M). The best results are highlighted in bold.**
> | Dataset | Method | M=0 MAE | M=0 MAPE | M=0 RMSE | M=0.3 MAE | M=0.3 MAPE | M=0.3 RMSE | M=0.5 MAE | M=0.5 MAPE | M=0.5 RMSE | M=0.7 MAE | M=0.7 MAPE | M=0.7 RMSE |
> | :--- | :--- | :---: | :---: | :---: | :---: | :---: | :---: | :---: | :---: | :---: | :---: | :---: | :---: |
> | **KnowAir** | Geo-FNO **[2]** | 22.50 | 0.61 | 34.88 | 24.25 | 0.70 | 38.66 | 26.62 | 0.78 | 42.38 | 28.20 | 0.86 | 43.83 |
> | | GNO **[1]** | 21.86 | 0.63 | 34.14 | 23.87 | 0.65 | 38.84 | **25.17** | 0.76 | 42.25 | 27.46 | 0.86 | 45.93 |
> | | Geo-FNO (standalone) **[2]** | 22.87 | 0.60 | 35.95 | 27.03 | 0.87 | 39.94 | 30.34 | 1.06 | 42.90 | 32.97 | 1.19 | 45.43 |
> | | GNO (standalone) **[1]** | 22.58 | 0.60 | 35.29 | 25.60 | 0.75 | 40.17 | 28.02 | 0.87 | 42.83 | 31.43 | 1.06 | 45.84 |
> | | **Our Model** | **20.55** | **0.55** | **33.05** | **22.89** | **0.65** | **36.63** | 25.20 | **0.74** | **39.84** | **27.19** | **0.82** | **42.58** |
> | **PEMS-BAY** | Graphormer **[4]** | 1.86 | 0.04 | 4.01 | 2.94 | 0.07 | 6.44 | 3.79 | 0.10 | 7.65 | 4.23 | 0.11 | 8.28 |
> | | STDEN **[5]**  | 2.39 | 0.06 | 4.95 | 3.91 | 0.09 | 7.31 | 4.92 | 0.11 | 8.58 | 7.46 | 0.16 | 11.57 |
> | | **Our Model** | **1.66** | **0.04** | **3.61** | **2.75** | **0.07** | **5.85** | **3.52** | **0.09** | **7.04** | **4.25** | **0.11** | **8.19** |
>
> ---
>
> ### 2. Writing quality and presentation
> We have refined the manuscript to improve clarity, consistency, and readability.
> 1.  **Notation updates:** Abstract notations such as $\hat{X}$ and $\tilde{X}$ have been replaced with concrete spectral-domain variables like $X_{\text{gft}}$ and $X_{\text{gtft}}$.
> 2.  **Standardization:** Mathematical symbols were standardized (e.g., unified notation $X_{1:T}$).
> 3.  **Visual clarity:** Key figures(Figure.2 and Figure.3) were redrawn and annotated so that intermediate variables are interpretable without referring to the main text.
>
> These modifications collectively improve precision and coherence.
>
> ---
> ### References
>
> **[1] GNO:** Anandkumar, Anima, et al. "Neural operator: Graph kernel network for partial differential equations." *ICLR 2020 workshop on integration of deep neural models and differential equations*. 2020.
>
> **[2] Geo-FNO:** Li, Zongyi, et al. "Fourier neural operator with learned deformations for pdes on general geometries." *Journal of Machine Learning Research* 24.388 (2023): 1-26.
>
> **[3] STGODE:** Fang, Zheng, et al. "Spatial-temporal graph ode networks for traffic flow forecasting." *Proceedings of the 27th ACM SIGKDD conference on knowledge discovery & data mining*. 2021.
>
> **[4] Graphormer:** Ying, Chengxuan, et al. "Do transformers really perform badly for graph representation?." *Advances in neural information processing systems* 34 (2021): 28877-28888.
>
> **[5] STDEN:** Ji, Jiahao, et al. "STDEN: Towards physics-guided neural networks for traffic flow prediction." *Proceedings of the AAAI conference on artificial intelligence*. Vol. 36. No. 4. 2022.

---

> ### Author Response · Authors · 2025-11-21
> **Part 2 of the Response (2/2)**
>
> ### 3. Visualization and interpretation of TGSSP(Time-Gated Spectral Segmentation Perception)
> We enhance TGSSP’s interpretability through a case study of its mode-wise scaling factors $\alpha_k$ and temporal gating activations, with all visualizations presented in the newly added "Spectral Specialization and Temporal Adaptation" subsection of **Section 4.4**.
>
> 1. **Spectral specialization:** Training trajectories of $\alpha_k$ reveal a competitive “spectral selection” effect: informative high-band modes are progressively amplified while less relevant modes decay, indicating learned, task-driven spectral weighting rather than uniform filtering.
>
> 2. **Frequency-adaptive gating:** During inference, low-frequency components adopt a near pass-through gating policy to preserve global trends, whereas high-frequency components show highly selective, high-variance gating that spikes around transient bursts and relaxes as signals stabilize, effectively acting as spectral-domain attention.
>
> 3. **Band-specific temporal focus:** Temporal kernel profiles differ systematically across frequency bands, showing that TGSSP learns band-dependent temporal receptive fields. This confirms that it jointly specializes in spectrum and time, enabling robust modeling of multi-scale, non-stationary dynamics.
>
> These observations offer a compact and transparent interpretation of how TGSSP coordinates spectral specialization with temporal adaptation to capture complex spatiotemporal behavior.
>
> ---
>
> ### 4. Model scalability on large graphs
> We clarify the computational characteristics of our model and the trade-offs involved in scaling to massive graphs.
> 1.  **Empirical efficiency:** Our model achieves state-of-the-art accuracy while requiring only $\sim$30% of the parameters and GPU memory compared to the next-best baseline. This demonstrates significant efficiency on standard benchmark datasets.
> 2.  **Validation on larger graphs:** We have successfully validated the model on the SD dataset (700+ nodes) from the **LargeST benchmark [6]**, where it maintains leading performance. Experiments on the full CA dataset **[6]** (containing over 5,000 nodes) are currently in progress and results will be reported upon completion.
> 3.  **Approximation trade-offs:** We investigated scalable extensions such as low-rank approximations or sparse spectral methods. However, we found that these mainstream approximation techniques (e.g., Nyström) tend to undermine our strategy of explicit spectral segmentation and independent temporal gating, leading to a loss in prediction precision. For instance, replacing the full GT-FNO with a Nyström approximation on **PEMS-BAY [7]** reduces the parameter count ($\sim$86K vs. $\sim$123K) but degrades the MAE from 1.66 to 1.96. This indicates that while the model can be compressed, preserving the full spectral resolution is critical for capturing the high-frequency dynamics that drive our superior performance.
>
> ---
>
> ### 5. Continuity modeling and graph spectral representations
>  As detailed in **Appendix A.2**, unlike traditional deep learning which maps finite-dimensional vectors, neural operators approximate the solution operator between continuous function spaces (a paradigm of "infinite-dimensional learning"). Consequently, learning the operator is mathematically equivalent to learning its integral kernel.
>
> Our graph spectral modeling is explicitly designed to efficiently and sufficiently learn this kernel. As elaborated in the "Continuous Reconstruction" paragraph of **Section 3.3**, the graph spectral basis offers an orthogonal and compact parameterization of the kernel directly on graph domains.
>
> This establishes that our spectral framework is a concrete realization of continuous operator learning principles.
>
> ---
>
> ### 6. Clarification of the truncation operation
> We have revised the manuscript to provide a more precise description of the truncation mechanism presented in Section 3.3 (Time-Gated Spectral Segmentation Perception).
> Truncation is performed _after_ applying both the GFT and the temporal FFT. Frequency thresholds divide the joint spectrum into low-, mid-, and high-frequency bands. Each band is independently processed in TGSSP.
> This multi-band decomposition yields a hierarchical representation that captures global trends, local variations, and transient phenomena, where the band boundaries follow established information-theoretic criteria.
>
> ---
> Yours sincerely,
>
> The authors of Paper 18391
> ### References
>
> **[6] LargeST (SD/CA):** Liu, Xu, et al. "Largest: A benchmark dataset for large-scale traffic forecasting." Advances in Neural Information Processing Systems 36 (2023): 75354-75371.
>
> **[7] PEMS-BAY:** Li, Yaguang, et al. "Diffusion convolutional recurrent neural network: Data-driven traffic forecasting." arXiv preprint arXiv:1707.01926 (2017).

---

> ### Author Response · Authors · 2025-11-26
> **Thanks for your valuable suggestions and Looking forward to further discussion**
>
> Dear Reviewer sBzk,
>
> We sincerely appreciate the thoughtful and detailed review you provided for our manuscript. Following your insightful comments, we have carefully prepared a point-by-point response addressing your concerns:
>
> **Part 1 of the Response (1/2):** We expand the baseline comparison to include recent neural-operator and physics-informed models (GNO, Geo-FNO, Graphormer, STDEN), provide unified results on KnowAir and PEMS-BAY, and improve writing quality and presentation through clearer notation, standardized symbols, and revised figures.
>
> **Part 1 of the Response (2/2):** We add visualization-based analyses to interpret TGSSP’s spectral specialization and temporal gating, clarify model scalability on large graphs with new empirical evidence and approximation studies, strengthen the link between continuous operator learning and graph spectral representations, and refine the explanation of the truncation-based multi-band decomposition.
>
> Thank you again for your time, constructive suggestions, and thoughtful evaluation. As the discussion phase is progressing, we would be truly grateful if you could kindly take a moment to engage further when convenient.
>
> ---
>
> Warm regards,
>
> Authors of Submission 18391

---

### Author Response · Authors · 2025-12-01
**Part 1 of Summary(1/2)**

**Dear Area Chair and reviewers**,

We sincerely thank the Area Chair and reviewers for their time and constructive feedback on our research paper entitled  *“Enabling arbitrary inference in spatio-temporal dynamic systems: A physics-inspired perspective”* .

**Paper summary**

Our paper tackles the long-standing challenge of predicting arbitrary spatiotemporal elements in dynamic systems. We propose **PhySTA**, where Continuous Spectrum–Temporal Learning (**CoSTL**) extends continuous-learning neural operators to graphs through a Graph–Time Fourier Neural Operator(**GT-FNO**) and a Time-Gated Spectral Segment Perception module for fine-grained spectral segmentation and dynamic temporal weighting. This enables prediction at arbitrary target nodes in continuous space, while Adaptive Multi-scale Interaction (**AMI**) further refines inference through multiscale node–edge interactions on a hierarchical graph. Experiments across multiple datasets demonstrate clear improvements in both effectiveness and efficiency over peer models.

**Reviewer-recognized contributions and strengths**

During the review process, this paper has received an overall positive assessment with an initial score of **5.33**. The reviewers highlighted several key strengths:

**Theoretical extension of continuous operators to graphs:**  The proposal of GT-FNO successfully extends Fourier Neural Operators to non-Euclidean, directed graphs via joint graph–time spectral decomposition, marking a significant conceptual advance in continuous spatiotemporal modeling. (Reviewer sBzk, Reviewer ucx5)

**Effective long-range graph modeling:**  The AMI module effectively captures coupled long-range dependencies within a single layer, addressing the over-smoothing and inefficiency issues common in deep GNNs. (Reviewer sBzk)

**High efficiency and scalability with SOTA performance:**  PhySTA achieves consistent SOTA performance with significantly lower overhead, offering an excellent trade-off between accuracy and scalability. (All Reviewers)

**Robustness under data sparsity:**  The model demonstrates exceptional robustness to data sparsity, maintaining top performance even under high node-masking ratios (arbitrary inference). (Reviewer ucx5)

**All concerns raised by reviewers have been fully addressed and we list them as below for your references.**

---

### 1. Comprehensive baseline coverage and SOTA validation

To ensure rigorous benchmarking, we expanded our evaluation scope. Although standard operators (GNO, Geo-FNO) are grid-based, we adapted them for graph structures per reviewer suggestions. We also evaluated Transformer/PDE-based methods (Graphormer, STDEN). Experiments on air quality and traffic datasets confirm GT-FNO’s superiority. Notably, substituting GT-FNO with these operators within our framework still yields competitive results, demonstrating the AMI module’s capability to correct and enhance operator predictions.

### 2. Deepened interpretability (Section 4.4 case study)

We added a detailed analysis, *Spectral Specialization and Temporal Adaptation*, explaining internal behavior of **Time-Gated Spectral Segment Perception**.

**Spectral selection:**  Training trajectories show that learnable scaling factors (α) progressively amplify informative high-frequency modes while suppressing noise.

**Dynamic gating:**  Inference-phase visualization reveals that temporal gates function as a spectral attention mechanism, synchronizing with transient signal bursts to maximize signal-to-noise contrast.

---

> ### Author Response · Authors · 2025-12-01
> **Part 2 of Summary(2/2)**
>
> ### 3. Verified scalability, efficiency, and design rationale
>
> We validated the model’s scalability and justified our architectural trade-offs through multi-dimensional analysis.
>
> **Scalability verification on large graphs:**  Extending our validation to the **GBA dataset (2,800+ nodes)** , PhySTA achieves a superior MAE of **23.55** with only **10.63 GB** of GPU memory, significantly surpassing DGCRN (25.49 MAE, 85.21 GB) and GWNet (28.14 MAE, 16.21 GB), demonstrating both robustness and resource efficiency on large-scale topologies.
>
> **Necessity of full spectral resolution:**  Low-rank approximations (e.g., Nyström) reduce parameters (123K → 86K) but substantially degrade accuracy (PEMS-BAY MAE 1.66 → 1.96), confirming that full spectral resolution is essential for capturing high-frequency dynamics.
>
> **AMI design logic (static vs. differentiable):**  Using offline Louvain clustering is intentional for stability. Urban macro-structures evolve slowly, so static partitioning efficiently captures physical community hierarchies without the instability and memory overhead of differentiable pooling, while rapid variations are handled by feature-driven interaction modules.
>
> ### 4. Theoretical and technical clarifications
>
> **Continuous operator learning:**  We clarified the connection between our method and infinite-dimensional operator learning theory, as originally detailed in **Appendix A.2** and **Section 3.3**. Specifically, we highlighted how the GT-FNO and AMI modules extend continuous function mapping to the graph domain, ensuring the theoretical underpinnings are explicitly linked to the proposed architecture.
>
> **Dynamic topology handling:** We clarified that although the adjacency matrix is fixed, PhySTA models dynamic topology through two implicit mechanisms. TGSSP uses spectral segmentation and temporal gating to infer topology-related variations from real-time node features, enabling frequency-domain approximation of evolving dynamics. Meanwhile, AMI introduces adaptivity through node–edge feature coupling and multi-scale subgraph interaction, reducing reliance on a single static topology.
>
> **Unseen node inference and prevention of information leakage:** We clarify that masked nodes are zero-padded at input and remain in the graph, so the model infers them purely through learned interaction patterns without accessing their labels. During testing, only features are masked to evaluate inference capability. As for multi-scale subgraphs, the coarse level abstracts cluster interactions, while the mid and fine levels retain original and enhanced long-range connectivity, allowing complementary recovery and preventing both information loss and leakage.
>
> **Presentation:**  We standardized notation (e.g., replace $\hat X$ and $\tilde X$ with explicit $X_{gft}$ and $X_{gtft}$) and redesigned Figures 2 and 3 with clearer annotations to improve readability.
>
> ---
>
> We believe these revisions comprehensively address the concerns raised. We remain committed to incorporating any further suggestions to ensure the final manuscript meets the highest standards.
>
>
> Warm regards,
>
> Authors of Submission 18391

---

### Meta-Review · Area_Chair_WGdL · 2026-01-06

**Summary:**

The paper proposes a physics-inspired framework PhySTA  for arbitrary inference in spatio-temporal graph systems by combining a graph-time Fourier neural operator (GT-FNO) with time-gated spectral segmentation (TGSSP) and a multi-scale node–edge interaction module (AMI) for refinement.

Reviewers’ main strengths were: (1) a meaningful extension of FNO-style operator learning to non-Euclidean graphs with clear architectural novelty, (2) strong robustness under high node-masking / sparsity and consistent performance on traffic + air-quality benchmarks, and (3) good efficiency trade-offs (parameters/memory/FLOPs) with supportive ablations. The rebuttal addressed key concerns effectively by adding missing baselines (operator/transformer/PDE-style), clarifying truncation/notation and improving figures, and providing interpretability (TGSSP scaling + gating visualizations) plus a clearer scalability discussion (including a Nyström approximation study and rationale for offline Louvain preprocessing).

Overall, the revised paper presents solid novelty and evidence for its claims, and the responses materially strengthen completeness and clarity; I recommend accept.

**Reviewer Concerns:**

Still outstanding concerns:
- Fundamental scalability limits of full graph spectral decomposition on very large (city-scale) graphs;
- Lack of an end-to-end differentiable alternative to the static multi-scale graph construction;

**Reviewer Scores:**

For Reviewer 5Bv2, I think the clarified theoretical link between spectral modeling and continuous operators, explicit truncation details, and improved notation resolve most raised weaknesses.

---

### Decision · Program_Chairs · 2026-01-26

Accept (Poster)